# Optimization Inspired Few-Shot Adaptation for Large Language Models

**Boyan Gao**[1⊠]     **Xin Wang**[1]     **Yibo Yang**[1,2⊠]     **David A. Clifton**[1,3]

[1] University of Oxford
[2] King Abdullah University of Science and Technology
[3] Oxford-Suzhou Institute of Advanced Research

## Abstract

Large Language Models (LLMs) have demonstrated remarkable performance in real-world applications. However, adapting LLMs to novel tasks via fine-tuning often requires substantial training data and computational resources that are impractical in few-shot scenarios. Existing approaches, such as In-context learning and Parameter-Efficient Fine-Tuning (PEFT), face key limitations: In-context learning introduces additional inference computational overhead with limited performance gains, while PEFT models are prone to overfitting on the few demonstration examples. In this work, we reinterpret the forward pass of LLMs as an optimization process, a sequence of preconditioned gradient descent steps refining internal representations. Based on this connection, we propose Optimization-Inspired Few-Shot Adaptation (OFA), integrating a parameterization that learns preconditioners without introducing additional trainable parameters, and an objective that improves optimization efficiency by learning preconditioners based on a convergence bound, while simultaneously steering the optimization path toward the flat local minimum. Our method overcomes both issues of ICL-based and PEFT-based methods, and demonstrates superior performance over the existing methods on a variety of few-shot adaptation tasks in experiments.

## 1   Introduction

The compelling performance of Large Language Model (LLM) has been demonstrated in real-world applications such as code generation [13, 43, 45], scientific reasoning [69, 14], healthcare [57, 66, 67], and robotics [10, 56]. This phenomenon can be attributed to the adaptation of pretrained base models toward the target tasks. Full parameter fine-tuning as a straightforward method requires tremendous computational resources and training data, which is usually not practical. Parameter-Efficient Fine-Tuning (PEFT) [29, 73, 46, 27] methods aim to reduce these costs by partially tuning the parameters, while these algorithms still require a relatively large amount of high-quality training data. Especially, when only a few data samples are given for adaptation to new tasks, they suffer from the overfitting problem and fail to learn generalizable adapters [41, 44, 31, 20, 37].

To enable adaptation with few-shot data on new tasks, In-context learning (ICL) [54, 11] offers an alternative approach by leveraging prompt engineering techniques. It stores a small set of demonstration examples in a buffer and modifies the forward pass to enable LLMs to generate answers for new queries. While ICL reduces data cost and mitigates the overfitting problem of parameter-efficient fine-tuning (PEFT), it still faces several significant challenges. For instance, the stored demonstration samples introduce additional computational burdens, slowing the inference

---

⊠ Corresponding authors: boyan.gao@eng.ox.ac.uk, yibo.yang93@gmail.com

39th Conference on Neural Information Processing Systems (NeurIPS 2025).

process. Besides, the improvement of the model on the target domain is highly constrained, since limited or even no learnable parameters are used for adaptation, resulting in the incapability of ICL algorithms to absorb the entire knowledge presented in the data and generalize to unseen data. When the demonstration examples exceed a certain threshold, the model's performance is usually saturated [39, 41]. In addition, the prompt format has an unpredictable impact on the ICL's performance [68, 81], and the existing mechanism designs are usually intuitive without theoretical support, leading to unexplainable failures. In this work, we address the following question:

*For few-shot adaptation, how can we develop an efficient method that avoids overfitting to few-shot data, as commonly observed in PEFT, while also overcoming ICL's lack of learnable parameters and extra inference cost?*

Existing works [64, 15, 4, 3, 8, 71, 79, 74] have demonstrated that the forward pass of an LLM for few-shot adaptation can be deemed as an optimization process with a sequence of gradient descent (GD) steps. However, these GD steps usually ignore the task-specific preconditioning matrices. As a result, this optimization process is not controllable, leading to sub-optimal adaptation performance. To this end, we first extend this process as preconditioned gradient descent (PGD), where the LayerNorm layers are integrated as learnable preconditioning matrices, which do not introduce learnable parameters while enabling the control of the few-shot adaptation process to avoid overfitting.

Thanks to our learnable preconditioners, we propose to steer the optimization trajectory toward task-specific solutions by enhancing two key properties: optimization efficiency and generalization ability. Since the number of optimization steps is tied to the number of attention layers, we first introduce an objective that promotes smoother optimization paths by optimizing step ratio, which implicitly tightens convergence bounds and improves optimization efficiency. To enhance generalization ability, we further propose an additional objective term that encourages convergence to flat regions of the loss landscape by minimizing the local sharpness. However, directly computing the sharpness is intractable. Our method estimates sharpness indirectly by minimizing the trace of the preconditioned Hessian at each step using the Hutchinson approximation [2]. As a result, unlike prior sharpness estimation approaches [23, 82, 32], often incurring significant computational overhead, our approximation makes it more scalable and LLM-compatible.

In summary, we introduce a novel optimization-inspired framework for few-shot adaptation, OFA, which improves both optimization efficiency and generalization ability for few-shot adaptation by steering the internal optimization via learnable preconditioners. It provides a new technical solution to this task, avoiding both issues of PEFT requiring expensive computational resources and adaptation datasets, and ICL relying on unstable prompt engineering techniques and extra inference cost. Extensive experiments across various datasets and LLM architectures demonstrate the superior performance of OFA over existing baselines. The contributions are listed as follows:

- We propose Optimization Inspired Few-Shot Adaptation (OFA), which frames the few-shot adaptation task as the learning of iteration-wise preconditioning matrices within the internal LLM optimization process, overcoming both issues of ICL-based and PEFT-based methods.

- We design the learning objectives to learn these internal optimization preconditioning matrices for enhancing the optimization efficiency and generalization ability while analyzing their contribution to the convergence speed and generalization bound theoretically.

- The proposed algorithm demonstrates superior performance among all the baseline models, including both ICL-based and PEFT, mainly LoRA-based, methods. Notably, OFA can achieve improvements of 4% - 10% with Llama2-7B and Llama3-8B-Instruct on all the challenging benchmarks compared with the SOTA method, I2CL [39].

## 2   Related Work

**Transformer implements gradient descent.** The recent works demonstrate that the pre-trained transformers, Large Language Models, can implement optimization algorithms such as gradient descent, with each attention layer corresponding to one optimization iteration [64, 15, 4, 3, 8, 71, 79, 72]. Without changing the parameters, LLMs can adapt to novel tasks with only a few demonstration examples through implicitly conducted optimization algorithms with similar behavior of multiple step gradient descent. This phenomenon has also been empirically observed in [15, 64]. Based on

this, one line of study [35] modifies the forward pass mechanism to improve the few-shot adaptation performance. Then the later research work explores the underlying property from a variety of perspectives, including the initialization, the demonstration sample efficiency [1], and complicated minmax optimization [30]. Ahn et al. [3] further claims that the preconditioned gradient descent algorithm can be learned on the random samples, whose preconditioning matrices vary according to the input feature distribution of the layer. Based on these studies, we aim to improve the optimization efficiency from the convergence speed and generalization perspective under the constraint that only a fixed number of certain optimization steps are accessible.

**Efficient model adaptation.** The pretrained models are expected to capture transferable knowledge for the benefit of novel task training efficiency on the computational resource and data samples. One line of research focuses on adapting models to the target tasks when a few samples are available [21, 48, 5, 60, 55, 58]. To achieve this, few-shot learners [21, 48] learns a set of transferable parameter initialization on the related tasks, thus with the limited number of training samples and adaptation steps, the model can converge to optimums. The following research works further extend this idea by developing advanced optimization geometry [51, 25, 26], learnable adaptation process components [38], and accurate gradient estimation [22]. Another lines of research explore a generalizable feature space to enable category separation by learning advanced metrics and the position of categories [58, 63, 6, 9, 75]. In the LLM era, adapting the pretrained model with low cost, namely the computational resources and the amount of data points, is in high demand. Parameter-efficient fine-tuning (PEFT) models reduce the adaptation spends by identifying the efficient tuning components, learning the row rank adapters [29, 42] and their initialization [73, 46, 27]. Even though these methods reduce adaptation cost dramatically in comparison with full model adaptation, they still fail to generalize when only a few samples are allowed. To the best of our knowledge, Liu et al. [40] shares a similar motivation to narrow the gap between PEFT and few-shot adaptation with ours; however, their work focuses on the empirical tricks and introduces extra parameters and increases the computational burden in the inference stage. In this work, we utilize the LLM property, that the inference process can be theoretically interpreted as the optimization process under the In-context learning region, and design novel objective terms to enable fast convergence and generalization.

## 3 Method

In this section, we introduce the proposed method for adapting the model using a few demonstration samples. Building on our insight that the optimization path, implicitly defined by the forward pass of a large language model (LLM), can be steered by modifying layer-wise preconditioning matrices, we propose Optimization Inspired Few-Shot Adaptation. Our method is designed to address two key essential properties for effective adaptation: optimization efficiency and generalization ability. These are encouraged through two corresponding penalty objective terms.

### 3.1 Optimization-inspired perspective for LLMs

The pretrained LLMs implement gradient descent for the adaptation to the target domain when prompted with the demonstration samples [64, 15, 4, 3, 8]. More formally, with $n$ query-answer prompt pairs, denoted as $x \in \mathbb{R}^d$ and $y \in \mathbb{R}$, the LLM model yields an answer $\hat{y}^{(n+1)}$ regarding the novel query $x^{(n+1)}$. We simplify the notations with matrix format by denoting $Z_i$ as the output from the $i$-th layer, while $Z_0$ is framed as the raw input data:

$$Z_0 = \begin{bmatrix} z^{(1)} & z^{(2)} & \dots & z^{(n)} & z^{(n+1)} \end{bmatrix} = \begin{bmatrix} x^{(1)} & x^{(2)} & \dots & x^{(n)} & x^{(n+1)} \\ y^{(1)} & y^{(2)} & \dots & y^{(n)} & 0 \end{bmatrix} \in \mathbb{R}^{(d+1)\times(n+1)}.$$
(1)

where $d$ and $n$ denote the input dimension and number of demonstration examples, respectively, and 0 represents the replaceable unknown variable corresponding to $x^{(n+1)}$. It has been theoretically substantiated [3, 64, 79] that the $t$-th attention layer of a transformer-based LLM, $F(\cdot) = f_T \circ \dots \circ f_t \circ f_1(\cdot)$, implements an iteration of gradient descent:

$$Z_{t+1} = Z_t - \eta P_t \nabla \mathcal{L}(Z_t) \tag{2}$$
$$\textbf{s.t. } f_t(Z_t) = -\eta P_t \nabla \mathcal{L}(Z_t) = \text{Attn}(Z_t),$$

with the objective defined by

$$\mathcal{L} = \|F(Z_0)_{[d+1, n+1]} - y^*\|_2^2,$$

where $\eta$ represents the learning rate and $P_t = I$ is an identical matrix which does not modify the update information, $\eta P_t \nabla \mathcal{L}(Z_t)$, implemented by an attention layer, $f_t(\cdot)$. As the ideal preconditioning matrix depends on the input data distribution [3], in this work, we learn the layer (iteration) wise preconditioning matrix, characterizing the task-specific optimization path.

## 3.2 Parameterization for Learnable Preconditioning Matrix

Building on the theoretical insight that an attention layer can be interpreted as a gradient descent (GD) step, we integrate learnable preconditioning matrices via LayerNorm, an often overlooked component in prior analytical works [24, 3]. Owing to its small parameter size and strategic position within the Transformer architecture, LayerNorm serves as a lightweight and tuning-efficient parameterization of the preconditioners for preconditioned GD. Specifically, in modern LLMs such as Llama [61, 62] and GPT-2 [53], each LayerNorm layer is parameterized by a single vector, resulting in fewer parameters than even a rank-1 LoRA model. These layers are typically placed after attention blocks and normalize the output of those blocks:

$$Z_{t+1} = Z_t - \Gamma_t \cdot \frac{\nabla \mathcal{L}(Z_t) - \mu_t}{\sigma_t}, \ \ \Gamma_t = \text{diag}(\gamma_t),$$

where $\mu_t$ and $\sigma$ are the mean and standard deviation of $\nabla \mathcal{L}(Z_t)$, and $\Gamma_t = \text{diag}(\gamma_t)$ represent the learnable diagonal matrix in the LayerNorm. Then the learnable preconditioning matrix in this optimization process is characterized as:

$$Z_{t+1} = Z_t - P_t \nabla \mathcal{L}(Z_t), \ \ P_t = \Gamma_t \cdot \frac{1}{\sigma_t},$$

where the potential bias term $\Gamma_t \cdot \frac{\mu_t}{\sigma_t}$ can be simplified as zero since in the modern LLMs [70, 61, 62, 52, 7, 77], LayerNorm is usually instantiated by RMSNorm [76] whose mean vector $\mu_t$ is set as zero vector instead of estimated during the forward pass.

## 3.3 Learning for Fast Convergence

By framing the forward pass of the transformer, fed with the prompt and query, the model gradually predicts the answer through an iterative optimization of the representation through the attention blocks. However, due to the architecture-specific constraints of LLMs, such as the fixed number of layers, it remains unclear whether the efficiency of this process is guaranteed or whether the process truly converges to an optimal solution.

To address these issues, we enhance optimization efficiency and stability by introducing a loss objective to mitigates the risk of gradient explosion and oscillation conducted by the forward pass. Specifically, by optimizing this loss objective, we refine the step ratios defined by:

$$\|Z_{t+1} - Z^*\| \leq \rho_t \|Z_t - Z^*\|, \ \ \rho_t < 1,$$

where $\rho_t$ works as a proxy reflecting the stability of the optimization process, and $Z^*$ denotes the optimum of the optimization problem. The new objective to equip this property for the few-shot adaptation by updating all the layer-wise preconditioning matrices, $P = \{P_t\}_{t=1}^T$, is defined by:

$$\mathcal{J}(P) = \sum_{t=1}^{T-1} \frac{\|Z_t - Z_{t+1}\|}{\|Z_t - Z_{t-1}\|}, \tag{3}$$

where we denotes the $l2$-norm by $\|\cdot\|$ through out the paper. One may notice that by decomposing the sum over all the layers, each term, $\frac{\|Z_t - Z_{t+1}\|}{\|Z_t - Z_{t-1}\|}$, increases the penalty strength when the numerator is larger than the denominator: $\|Z_t - Z_{t+1}\| > \|Z_t - Z_{t-1}\|$ as when $\|Z_t - Z_{t+1}\| \gg \|Z_t - Z_{t-1}\|$ indicates exploding or oscillating steps, suggesting poor conditioning or overshooting; When over minimizing the numerator in $\frac{\|Z_t - Z_{t+1}\|}{\|Z_t - Z_{t-1}\|}$ will be regulated by the denominator in $\frac{\|Z_{t+1} - Z_{t+2}\|}{\|Z_{t+1} - Z_t\|}$ and $\|Z_t - Z_{t+1}\| \ll \|Z_t - Z_{t-1}\|$ represents contraction, an indicator of convergence. Beyond enhancing the step-wise optimization quality, Eq. 3 also plays a crucial role in accelerating convergence, which we substantiate through analysis:

**Theorem 3.1.** *Let $f : \mathbb{R}^d \to \mathbb{R}$ be a twice continuously differentiable function with locally Lipschitz gradients. Suppose the update rule is given by:*

$$Z_{t+1} = Z_t - P_t \nabla \mathcal{L}(Z_t),$$

*where each $P_t \in \mathbb{R}^d \times \mathbb{R}^d$ is a learnable preconditioning matrix. Define the step-ratio objective in Eq. 3 Under the assumption that $f$ admits a local second-order Taylor expansion approximation at each step, then minimizing $\mathcal{J}(P)$ encourages the learned preconditioners $P_t$ to induce local operators $I - \eta P_t H_t$ with $H_t = \nabla^2 f(Z_t)$ with a smaller spectral radius.*

$$\|Z_{t+1} - Z^*\| \leq \rho_t \|Z_t - Z^*\|, \ \rho_t < \rho_{t-1}.$$

*Thus, it induces faster local contraction and improved convergence.*

The step-ratio objective $\mathcal{J}(P)$ serves as a differentiable proxy that captures the stability and efficiency of this optimization process. Smaller step ratios imply smoother convergence and discourage overshooting or oscillation. By optimizing $\mathcal{J}(P)$ over preconditioner parameters, we shape the feed-forward dynamics to mimic efficient optimization, inducing faster adaptation and better generalization in downstream tasks.

## 3.4 Learning for Flat Region Convergence

The effectiveness of the flat local minimum for the model's generalization ability has been theoretically and empirically explored. Motivated by this, we aim to enable the preconditioning matrix to be used for flatness-seeking ability by minimizing the sharpness of the loss landscape during the optimization process. However, the existing method for sharpness estimation [23, 32, 78] developed for the optimization process requires the explicit expression, while in our setting, such information is not accessible due to the black box characterization of the update information. In addition, those methods do not consider the effect of the local sharpness approximation from the preconditioning matrix. To handle this, we estimate sharpness for the layer-wise preconditioned GD optimization by the preconditioning Hessian trace:

$$\mathcal{H}_P = tr(P_t \nabla^2 \mathcal{L}(Z_t) P_t^T).$$

However, directly computing this trace is infeasible due to the implicitly defined optimization process, including the loss function and the gradients. Instead, we utilize a numerical method, the Hutchinson approximation [2]:

$$tr(P_t \nabla^2 \mathcal{L}(Z_t) P_t^T) \approx \frac{1}{\epsilon} \mathbb{E}_\nu \left[ \nu^T P_t (\nabla \mathcal{L}(Z_t + \epsilon P_t \nu) - \nabla \mathcal{L}(Z_t)) \right]$$

$$\approx \frac{1}{\epsilon} \frac{1}{N} \sum_i \left[ \nu_i^T P_t (\nabla \mathcal{L}(Z_t + \epsilon P_t \nu_i) - \nabla \mathcal{L}(Z_t)) \right], \tag{4}$$

where $\nu \sim \mathcal{N}(0, I)$ is a small perturbation sampled layer-wisely, and $tr(\cdot)$ represents the operator for trace calculation, $\epsilon$ denotes a small scale number. Note that in the non-convex optimization setting, $tr(P_t \nabla^2 \mathcal{L}(Z_t) P_t^T)$ can be negative. This may destabilize the training due to the numerical issue in the minimization process. To mitigate this issue and maintain the valuable information contained in the negative values, we regularize this term by adding a Softplus[18] activation function, $\delta(\cdot)$, to stabilize the numerical optimization while retaining the information brought by the negative trace. We provide the implementation details in Algorithm 1. We also analyze the connection between the flatness of the layer-wise preconditioning matrix and the generalization to understand the reason for the enhanced generalization ability.

**Theorem 3.2.** *Let $Z_T$ be the final parameters after $T$ steps of optimization, with preconditioning update rules in Eq. 2 and denoting $\nabla^2 \mathcal{L}_{train}(Z_t)$ as the Hessian at step $t$ with $\|P_t \nabla^2 \mathcal{L}_{train}(Z_t)\|_F$ measuring the curvature after preconditioning at that step. Assume the loss is smooth, $\|\nabla^2 \mathcal{L}(Z_t)\|_F \leq \mu$, and the gradient is bounded, $\|\nabla \mathcal{L}(Z_t)\| \leq G$, the generalization gap satisfies:*

$$\mathbb{E}[\mathcal{L}_{test}(Z_T) - \mathcal{L}_{train}(Z_T)] \leq \mathcal{O}\left( \sqrt{\frac{1}{n} \sum_{t=1}^{T} \|P_t \nabla^2 \mathcal{L}_{train}(Z_t)\|_F^2} \right).$$

---

**Algorithm 1** Sharpness estimation in Optimization Inspired Few-Shot Adaptation

---

1: **Input:** Input prompt: $Z_0$, Learnable preconditioners: $\{P_t\}_t$, Noise scale : $\epsilon$, and, Transformer: $\{f_t\}_t$
2: **Output:** $\{tr(P_t \nabla^2 \mathcal{L}(Z_t) P_t^T)\}_t$
3: The first forward pass: set $t = 0$
4: **while** $t < T - 1$ **do**
5:   $Z_{t+1} = f(Z_t)$
6:   $P_t \nabla \mathcal{L}(Z_t) = Z_{t+1} - Z_t$
7:   **for** $i$ in range($N$) **do**
8:     $\nu_i \sim \mathcal{N}(0, I)$
9:     $\hat{Z}_{t+1}^i = f_t(Z_t + \epsilon P_t v)$
10:     $P_t \nabla \mathcal{L}(Z_t + \epsilon P_t \nu_i) = \hat{Z}_{t+1}^i - (Z_t + \epsilon P_t v)$
11:   **end for**
12:   $tr(P_t \nabla^2 \mathcal{L}(Z_t) P_t^T) = Eq.\ 4$
13:   $t\mathrel{+}= 1$
14: **end while**

---

More intuitively, seeking the right preconditioning matrix at each step helps the optimizer follow the low-curvature valleys of the loss landscape, leading to solutions that are not only low-loss but also robust to perturbations, which is beneficial for generalization, and proof is given in Appx. C.

Building on the two theoretical results, we introduce two penalty terms into the preconditioner learning objective to guide inference features toward faster convergence in flatter regions of the loss landscape:

$$\Psi(P) = l_{CE}(F(Z_0)) + \lambda_1 \sum_{t=1}^{T-1} \frac{\|Z_t - Z_{t+1}\|}{\|Z_t - Z_{t-1}\|} + \lambda_2 \sum_{t=1}^{T-1} \delta(tr(P_t \nabla^2 \mathcal{L}(Z_t) P_t^T)), \qquad (5)$$

where $\lambda_1$ and $\lambda_2$ are the tunable hyperparameters, controlling the regularization strength for the convergence and local flatness, and $l_{CE}$ denotes CrossEntropy to guarantee the features, $Z_t$, are optimized towards the task-specific local minimum.

## 4 Experiments

In this section, we demonstrate the generalization ability of the calibrated Large Language Models on various settings. We begin by briefing the configuration of the experiments, including the architecture, datasets, and baseline models. We then dive into the efficiency of the contribution of the improvement of each proposed learning objective component.

**Tasks.** We follow the evaluation protocol utilised in [39], and apply the same tasks to evaluate Optimization Inspired Few-Shot Adaptation, which includes sentiment analysis: SST-2 [59], emotion classification: Emoc [12], question classification: TREC [36], topic classification AGNews [80], encompassing 5-way sentiment analysis: SST-5 [59], movie review classification: MR [50], 14-way topic classification: DBPedia [33], subjectivity status categorization: Subj [49], and the hate speech detection: HateSp18 [16]. All the datasets are downloaded from HuggingFace without further modification.

**Baseline Algorithms.** To evaluate OFA, we conduct comparisons with other methods sharing a similar motivation and are capable of consuming the demonstration samples along with the standard zero-shot and few-shot (ICL) baselines. We select the recent representative methods solving the tasks of interest from various directions to demonstrate the superior performance of OFA. **Soft-prompt** [34] learns a small set of continuous vectors prepended to the input of data to guide the model's behavior to a specific task. **Label-anchor** [65] shares a similar idea, aiming to learn with Soft-prompt methods, whereas learning the class label in the embedding space for few-shot or zero-shot adaptation. **Task-vector** [28] extracts the task representative vectors from the demonstration samples and injects them into the novel inner mechanism to steer the inference process, achieving the zero-shot complexity. **I2CL** [39] a recent state-of-the-art task-vector based method utilizing the residual stream property to eliminate the model-specific layer selection process. **IA3** [41] handles the limited adaptation sample

Table 1: Comparison between OFA and other baseline algorithms on Llama2-7B and Llama3-8B-Instruct. Mean accuracy and standard deviation across five random seeds are reported. **Best** results are highlighted in bold.

| Dataset | SST-2 | SST-5 | TREC | AGNews | Subj | HateSp18 | DBPedia | EmoC | MR |
|---|---|---|---|---|---|---|---|---|---|
| Method | | | | | Llama2-7B | | | | |
| Zero-shot | 83.00 | 27.00 | 50.00 | 70.20 | 51.40 | 54.20 | 72.00 | 41.80 | 73.60 |
| Few-shot (ICL) | $94.44_{\pm1.44}$ | $41.72_{\pm3.68}$ | $77.32_{\pm4.41}$ | $85.68_{\pm2.00}$ | $52.56_{\pm3.09}$ | $70.24_{\pm5.80}$ | $96.64_{\pm0.48}$ | $75.48_{\pm1.63}$ | $93.24_{\pm0.50}$ |
| Soft-prompt | $56.24_{\pm6.99}$ | $24.24_{\pm2.96}$ | $55.20_{\pm4.14}$ | $78.00_{\pm7.60}$ | $57.40_{\pm4.93}$ | $59.56_{\pm6.96}$ | $74.40_{\pm6.43}$ | $35.08_{\pm5.29}$ | $54.32_{\pm1.76}$ |
| Label-anchor | $83.32_{\pm5.95}$ | $27.68_{\pm4.21}$ | $77.48_{\pm3.49}$ | $83.72_{\pm1.04}$ | $53.00_{\pm2.95}$ | $64.52_{\pm8.09}$ | $81.40_{\pm3.67}$ | $59.12_{\pm10.60}$ | $84.40_{\pm5.89}$ |
| Task-vector | $81.44_{\pm4.73}$ | $25.96_{\pm0.59}$ | $65.68_{\pm1.93}$ | $79.68_{\pm4.07}$ | $58.56_{\pm4.91}$ | $67.68_{\pm3.70}$ | $89.48_{\pm2.58}$ | $44.64_{\pm3.53}$ | $82.32_{\pm5.37}$ |
| IA3 | $93.28_{\pm2.29}$ | $46.08_{\pm2.11}$ | $84.40_{\pm5.99}$ | $87.04_{\pm1.97}$ | $71.92_{\pm8.08}$ | $72.44_{\pm2.59}$ | $94.68_{\pm1.09}$ | $64.32_{\pm1.95}$ | $88.80_{\pm2.28}$ |
| I2CL | $87.68_{\pm2.47}$ | $39.12_{\pm2.69}$ | $78.56_{\pm5.32}$ | $85.48_{\pm1.16}$ | $73.84_{\pm3.84}$ | $69.88_{\pm5.67}$ | $90.16_{\pm1.86}$ | $63.72_{\pm1.37}$ | $87.68_{\pm2.26}$ |
| **OFA (Ours)** | $\mathbf{95.84}_{\pm0.41}$ | $\mathbf{50.36}_{\pm3.28}$ | $\mathbf{85.92}_{\pm1.90}$ | $\mathbf{89.00}_{\pm1.26}$ | $\mathbf{88.40}_{\pm4.76}$ | $\mathbf{83.04}_{\pm3.72}$ | $\mathbf{97.72}_{\pm0.52}$ | $\mathbf{76.60}_{\pm2.39}$ | $\mathbf{94.36}_{\pm1.13}$ |
| | | | | | Llama3-8B-Instruct | | | | |
| Zero-shot | 93.00 | 35.80 | 71.00 | 80.40 | 50.80 | 67.80 | 67.40 | 53.60 | 86.40 |
| Few-shot (ICL) | $96.48_{\pm0.48}$ | $46.72_{\pm2.64}$ | $79.92_{\pm5.83}$ | $89.64_{\pm0.59}$ | $57.48_{\pm7.08}$ | $52.72_{\pm2.35}$ | $97.00_{\pm0.28}$ | $65.28_{\pm4.29}$ | $93.12_{\pm0.16}$ |
| Soft-prompt | $84.68_{\pm7.71}$ | $38.40_{\pm5.68}$ | $75.68_{\pm8.17}$ | $84.96_{\pm3.80}$ | $73.28_{\pm5.41}$ | $62.72_{\pm5.54}$ | $82.88_{\pm6.45}$ | $55.32_{\pm9.74}$ | $75.76_{\pm7.71}$ |
| Label-anchor | $93.36_{\pm2.39}$ | $40.54_{\pm5.44}$ | $78.28_{\pm4.07}$ | $84.64_{\pm1.61}$ | $54.16_{\pm2.25}$ | $69.48_{\pm5.43}$ | $87.48_{\pm3.04}$ | $59.36_{\pm2.48}$ | $88.20_{\pm3.69}$ |
| Task-vector | $94.80_{\pm2.02}$ | $56.42_{\pm1.15}$ | $79.83_{\pm1.52}$ | $89.21_{\pm0.58}$ | $76.08_{\pm1.23}$ | $67.12_{\pm0.32}$ | $79.52_{\pm1.84}$ | $57.96_{\pm4.59}$ | $86.52_{\pm0.64}$ |
| IA3 | $94.32_{\pm0.82}$ | $49.24_{\pm2.06}$ | $87.60_{\pm3.46}$ | $88.36_{\pm1.80}$ | $82.04_{\pm7.43}$ | $77.20_{\pm4.37}$ | $92.56_{\pm1.82}$ | $68.04_{\pm2.24}$ | $91.76_{\pm0.43}$ |
| I2CL | $90.84_{\pm0.98}$ | $48.96_{\pm2.48}$ | $79.60_{\pm6.22}$ | $88.96_{\pm2.03}$ | $81.48_{\pm4.68}$ | $65.88_{\pm3.61}$ | $91.20_{\pm2.03}$ | $64.32_{\pm2.05}$ | $88.88_{\pm0.61}$ |
| **OFA (Ours)** | $\mathbf{97.08}_{\pm0.27}$ | $\mathbf{58.32}_{\pm2.74}$ | $\mathbf{89.06}_{\pm1.49}$ | $\mathbf{91.84}_{\pm0.61}$ | $\mathbf{92.64}_{\pm3.43}$ | $\mathbf{89.47}_{\pm0.47}$ | $\mathbf{97.92}_{\pm1.06}$ | $\mathbf{79.24}_{\pm4.87}$ | $\mathbf{94.56}_{\pm0.51}$ |

issue by reducing the trainable parameters while regularizing high probability on wrong predictions and accounting for the length of different answer choices.

**Main Results.** We compare OFA with baseline methods on four main decoder-only architectures: Llama2-7B, Llama3-8B, Llama3-8B-Instruct, and GPT2-XL. These architectures are selected for their suitable memory cost relative to our computational cost. We present the performance of OFA on Llama2-7B, and Llama3-8B-Instruct in Table 1, in which we can notice that OFA outperforms all the competitors across all the datasets with noticeable margins. Especially, on DBPedia and Subj, OFA demonstrates dramatic improvements. In the context that an attention layer performs an optimization step, we can observe that by retaining the main gradient part intact, tuning the preconditioning matrices is sufficient to improve the optimization efficiency. We leave the results of other models in Appx. A where a similar performance pattern can be observed.

**Ablation Study via Probe Analysis.** We study the per-layer feature quality generated by OFA via probing. To do this, we collected the training datasets by generating per-layer features by feeding the few-shot adaptation sets to the (trained) model and attaching the corresponding labels, then a set of linear classifiers is trained to predict the objects based on those features. For a fair comparison, the same process, including dataset collection and model training, is repeated on the raw model to construct the baseline. The learned classifiers are employed to predict the per-layer features yielded from the test data. To illustrate the effect of OFA, we plot the layer-wise accuracy and loss in Figure 1, from which one can observe that the model trained by OFA consistently outperforms the baseline model under both metrics across various datasets. More importantly, from an optimization dynamic perspective, the loss learning curve generated by OFA converges to a more stable region with the smallest fluctuation in comparison with other methods across different datasets, which indicates the flat convergence region. As preconditioning matrices steer the optimization path, directly comparing the steps for achieving the final loss could be unfair; however, we can still observe that OFA reaches the same loss level with fewer steps in Figure 1. Therefore, OFA not only provides a flat minimum but also improves the optimization efficiency.

**Layer-wise Sharpness Analysis.** We study the effect of OFA on the models' layer-wise behavior across different datasets. By estimating the average sharpness over different test samples by Eq. 4. One can observe that in Figure 2, the model trained by OFA consistently illustrates the lowest sharpness among the baseline models across all the layers. Especially, at the end of the optimization steps, without the regularization in Eq. 4, the sharpness quantity of the model trained by CE and the base model increases dramatically. This phenomenon reflects the sensitivity of the loss to the different test samples and determines the generalization performance of the model, which can be further justified by Table 1. To be more specific, the models attain an increase in sharpness at the final hidden layers, resulting in inferior test accuracy from the target domain.

**Layer-wise Step Ratio Analysis.** We evaluate the optimization quality of OFA by comparing the average step ratio on the test set across different optimization steps. Due to minimal visual differences

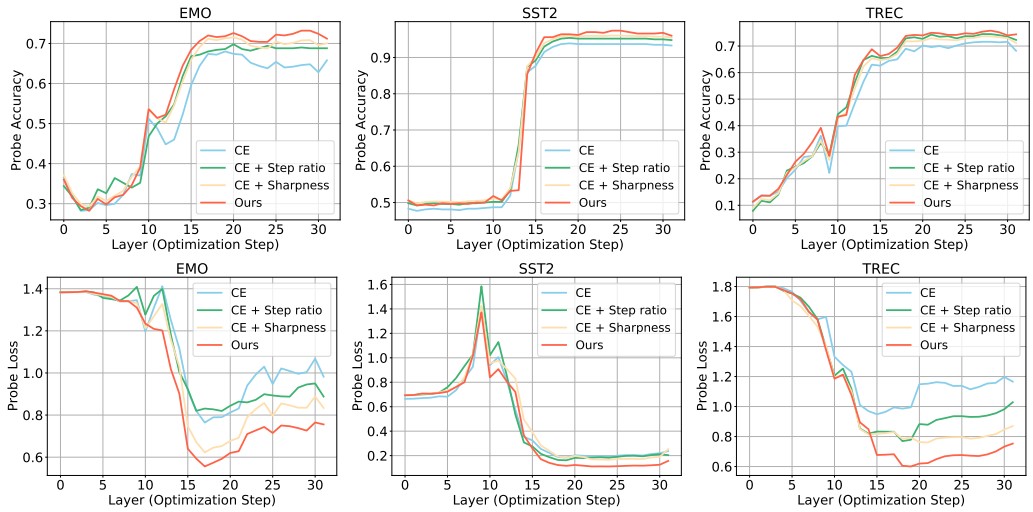

Figure 1: Probe Analysis on EMO, SST, and TREC. The layer-wise prediction accuracy (%) and loss on the test set comparison is conducted with four competitors, CE, CE + Step ratio, CE + Sharpness, and Ours. CE denotes the Llama2-7B model adapted to the target set through CrossEntropy loss via updating the layernorm parameters; CE + Step ratio follows the same adaptation protocol as CE but with Step ratio penalty attached in Eq. 3; CE + Sharpness uses Sharpness in Eq. 4 instead while Ours utilizing the OFA objective in Eq. 5.

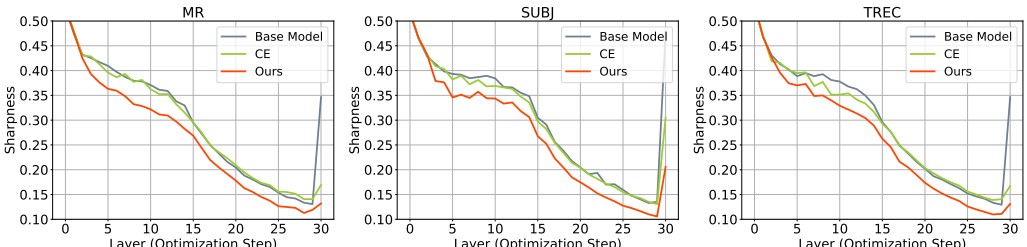

Figure 2: Sharpness comparison on MR, Subj, and TREC. The average sharpness over the test samples across different layers on three models, with base model denoting the few-shot (ICL) setting, CE representing the model trained by the CrossEntropy on the demonstration samples, and Ours trained by OFA via the same adaptation protocol as that utilized in CE.

in earlier layers, we focus on the last 16 layers in Figure 3. Notably, optimizing the step-ratio objective in OFA results in smoother and more consistent contraction across layers, highlighting the effectiveness of our learned preconditioning mechanism. In contrast, baseline models exhibit higher and more erratic step ratios, particularly with sharp increases in the later layers, suggesting an unstable optimization trajectory. Empirically, it is observed that models with flatter or more contractive step-ratio profiles tend to achieve better performance, supporting our analysis that step-ratio minimization enhances optimization efficiency. In addition, the step ratio is typically assumed to be smaller than one to ensure optimization convergence. However, as shown in Figure 3, particularly in the few-shot learning setting (Base Model) and in models trained with CE, this assumption does not empirically hold in the internal optimization dynamics of LLMs. Nevertheless, OFA remains robust to such violations and achieves faster convergence even with smaller step ratios. In other words, a step ratio smaller than one is not a strict requirement for our algorithm. Instead, OFA flexibly adapts to both compliant and non-compliant scenarios, optimizing the step ratio rather than imposing it.

**Comparison with LoRA.** We compare our method with LoRA [29] for the adaptation efficiency based on Llama2-7B [61]. A lightweight version where the learnable adapters are only applied to the value and query project layers is applied to different numbers of ranks, ranging from 1, 16, 64, and 128, to eliminate the effects from this hyperparameter selection. To further reduce the amount of learnable parameters, the bias sets of the adapters are not learned. The LoRA adapters are

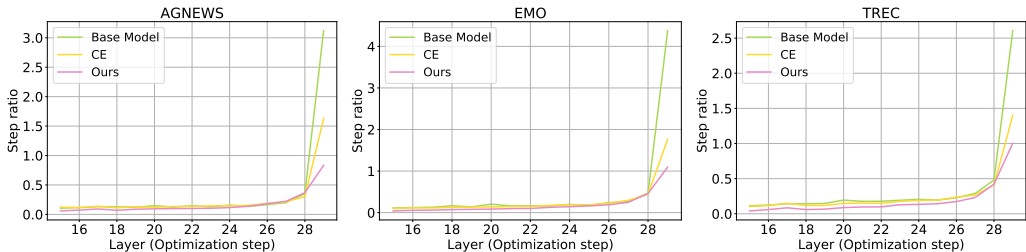

Figure 3: Step ratio comparison across the test sets of AGNews, Subj, and TREC over each layer of models based on Llama-7B. We compare the base model with demonstration examples (Base model), the model fine-tuned using CrossEntropy (CE), and the model tuned with OFA (Ours).

Table 2: The comparison between our method and LoRA on various datasets. Llama2-7B and Llama3-8B-Instruct are used as the base model, with ranks ranging from 1, 16, 64, and 128. All the methods are trained and evaluated with 5 trials with different random seeds, along with the mean performance on classification accuracy (%) and variance reported. The number of trainable parameters for all settings is attached.

| Dataset | Llama2-7B | | | | | |
|---|---|---|---|---|---|---|
| | Rank 128 | Rank 64 | Rank 16 | Rank 1 | Rank1 (our loss) | Ours |
| SST-2 | $87.64_{\pm 5.63}$ | $80.64_{\pm 15.38}$ | $86.36_{\pm 6.99}$ | $89.64_{\pm 3.23}$ | $88.48_{\pm 3.34}$ | $\mathbf{95.84}_{\pm 0.41}$ |
| SST-5 | $28.12_{\pm 9.20}$ | $37.16_{\pm 8.59}$ | $31.60_{\pm 9.10}$ | $24.84_{\pm 9.92}$ | $20.80_{\pm 0.89}$ | $\mathbf{50.36}_{\pm 3.28}$ |
| TREC | $52.60_{\pm 24.63}$ | $62.68_{\pm 21.30}$ | $33.68_{\pm 23.15}$ | $22.88_{\pm 9.09}$ | $24.32_{\pm 6.02}$ | $\mathbf{85.92}_{\pm 1.90}$ |
| AGNews | $82.40_{\pm 4.74}$ | $62.4_{\pm 26.26}$ | $73.16_{\pm 23.16}$ | $50.56_{\pm 31.36}$ | $62.04_{\pm 29.5}$ | $\mathbf{89.00}_{\pm 1.26}$ |
| Subj | $75.44_{\pm 9.57}$ | $70.84_{\pm 10.41}$ | $72.16_{\pm 14.52}$ | $72.08_{\pm 8.74}$ | $72.84_{\pm 11.33}$ | $\mathbf{88.40}_{\pm 4.76}$ |
| HateSpeech18 | $72.28_{\pm 10.41}$ | $73.88_{\pm 6.46}$ | $67.96_{\pm 12.51}$ | $69.14_{\pm 9.76}$ | $69.38_{\pm 10.77}$ | $\mathbf{83.04}_{\pm 3.72}$ |
| DBPedia | $93.20_{\pm 2.32}$ | $90.76_{\pm 3.60}$ | $95.16_{\pm 0.43}$ | $59.44_{\pm 42.01}$ | $74.6_{\pm 33.23}$ | $\mathbf{97.72}_{\pm 0.52}$ |
| EmoC | $34.40_{\pm 18.45}$ | $42.64_{\pm 21.86}$ | $58.96_{\pm 17.70}$ | $25.64_{\pm 2.95}$ | $33.24_{\pm 18.28}$ | $\mathbf{76.60}_{\pm 2.39}$ |
| MR | $82.68_{\pm 5.98}$ | $65.36_{\pm 18.40}$ | $74.12_{\pm 20.56}$ | $64.84_{\pm 13.86}$ | $64.88_{\pm 18.48}$ | $\mathbf{94.36}_{\pm 1.13}$ |
| Trainable parameters (Million) | 67.10 M | 33.55 M | 8.39 M | 0.53 M | 0.53 M | **0.27 M** |
| Dataset | Llama3-8B-Instruct | | | | | |
| SST-2 | $78.72_{\pm 13.37}$ | $88.32_{\pm 2.57}$ | $80.92_{\pm 12.05}$ | $87.08_{\pm 4.81}$ | $87.40_{\pm 8.05}$ | $\mathbf{97.08}_{\pm 0.27}$ |
| SST-5 | $27.80_{\pm 9.24}$ | $20.32_{\pm 1.17}$ | $27.76_{\pm 5.46}$ | $19.52_{\pm 0.45}$ | $20.32_{\pm 1.72}$ | $\mathbf{58.32}_{\pm 2.74}$ |
| TREC | $61.12_{\pm 28.41}$ | $59.00_{\pm 29.23}$ | $70.92_{\pm 30.32}$ | $25.88_{\pm 9.17}$ | $27.52_{\pm 6.09}$ | $\mathbf{89.06}_{\pm 1.49}$ |
| AGNews | $50.76_{\pm 26.15}$ | $50.76_{\pm 23.46}$ | $47.88_{\pm 24.60}$ | $39.72_{\pm 25.1}$ | $39.12_{\pm 24.40}$ | $\mathbf{91.84}_{\pm 0.61}$ |
| Subj | $77.84_{\pm 13.73}$ | $80.28_{\pm 6.97}$ | $81.92_{\pm 6.57}$ | $78.92_{\pm 8.43}$ | $80.96_{\pm 5.55}$ | $\mathbf{92.64}_{\pm 3.43}$ |
| HateSpeech18 | $71.08_{\pm 11.19}$ | $70.08_{\pm 9.56}$ | $69.36_{\pm 6.94}$ | $63.60_{\pm 9.07}$ | $68.92_{\pm 8.33}$ | $\mathbf{89.47}_{\pm 0.47}$ |
| DBPedia | $91.00_{\pm 1.07}$ | $88.32_{\pm 0.83}$ | $92.92_{\pm 2.07}$ | $57.88_{\pm 39.37}$ | $73.52_{\pm 32.69}$ | $\mathbf{97.92}_{\pm 1.06}$ |
| EmoC | $33.24_{\pm 13.28}$ | $38.44_{\pm 11.79}$ | $47.6_{\pm 17.84}$ | $24.68_{\pm 1.17}$ | $27.68_{\pm 3.65}$ | $\mathbf{79.24}_{\pm 4.87}$ |
| MR | $87.52_{\pm 2.34}$ | $87.60_{\pm 3.24}$ | $88.28_{\pm 1.95}$ | $87.20_{\pm 3.38}$ | $87.84_{\pm 13.59}$ | $\mathbf{94.56}_{\pm 0.51}$ |
| Trainable parameters (Million) | 54.53 M | 27.26 M | 6.82 M | 0.43 M | 0.43 M | **0.27 M** |

trained on the same adaptation datasets with the fairly tuned hyperparameter following the details in Appx. E. From Table 2, one can observe that OFA can defeat all the LoRA models, demonstrating a significant parameter efficiency for the adaptation with the few-shot demonstration examples, while the LoRA models, with the smallest amount of learnable parameters, still approximately double ours and struggle to achieve the same level of performance as ours. In addition, the LoRA rank is sensitive to the datasets, leading to a greater hyperparameter tuning burden, while in this very few sample case, LoRA models in general gain relatively high variance due to the overfitting on the demonstration sample selection. We trained a LoRA model with a similar parameter amount to our model, and our objective resulted in that OFA boosts the LoRA model performance but still fails to defeat ours. This is because the LoRA model dramatically modifies the essential optimization component, the gradient, while ours only tunes the preconditioning matrices.

**Sensitivity Analysis to Training Samples.** We analyze the impact of the number of training samples on the performance of OFA through experiments conducted on different models, Llama2 and Llama3-8B-Instruct, and datasets, HateSP18 and Subj. As shown in Table 3, the performance of OFA steadily improves as the number of training samples per category increases, eventually saturating when this number reaches 15. In contrast, the baseline model, ICL, exhibits higher sensitivity to changes in the

Table 3: Performance (Accuracy %) comparison on various shot samples. We illustrate the effects of the different few-shot samples, ranging from 3, 5, 10, 15, and 20, on ICL and OFA with Llama2-7B and Llama3-8B-Instruct. The prompt template used for ICL is provided in Appx. 6.

| Dataset | Method | Llama2-7B | | | | |
| | | 3-shot | 5-shot | 10-shot | 15-shot | 20-shot |
|---|---|---|---|---|---|---|
| HateSP18 | Few-shot (ICL) | $68.81_{\pm4.61}$ | $70.24_{\pm5.80}$ | $70.64_{\pm5.64}$ | $74.23_{\pm6.72}$ | $70.72_{\pm5.93}$ |
| | OFA | $77.63_{\pm4.72}$ | $83.04_{\pm3.72}$ | $83.73_{\pm3.11}$ | $86.09_{\pm3.59}$ | $84.97_{\pm4.73}$ |
| | | Llama3-8B-Instruct | | | | |
| Subj | Few-shot (ICL) | $63.76_{\pm6.93}$ | $57.48_{\pm7.08}$ | $54.73_{\pm6.46}$ | $61.80_{\pm5.84}$ | $58.32_{\pm5.62}$ |
| | OFA | $88.76_{\pm5.80}$ | $92.64_{\pm3.43}$ | $93.62_{\pm4.12}$ | $95.40_{\pm3.85}$ | $95.23_{\pm4.24}$ |

Table 4: Model complexity comparison. We compare the theoretical inference parameter complexity introduced by the ICL-based methods with OFA where M, D, and L represent the number of demonstration tokens, the model's dimensionality, and the number of layers in the architecture, respectively. Q denotes the number of additional learnable tokens used in the Soft-prompt method, while 1/K corresponds to the compression rate of the associated context-compression technique. We also attach the practical average time (seconds) cost on DBPedia, the most time-consuming one, over five trials.

| Dataset | Zero-shot | Few-shot (ICL) | Soft-prompt | Label-anchor | Task-vector | I2CL | OFA |
|---|---|---|---|---|---|---|---|
| Introduced parameters | 0 | 2MDL | 2DL | (2M+Q)DL | 2(M/K)DL | 2DL | 0 |
| Inference cost (s) | 51.24 | 59.93 | 53.64 | 52.41 | 56.78 | 52.59 | 51.37 |

number of training samples, showing larger performance variance and failing to match the consistent performance of OFA.

**Inference Cost.** We compare the inference-time computational complexity of our model against baseline methods in Table 4. Notably, since OFA is designed to adapt to the target domain at inference without additional overhead, it introduces no theoretical increase in computational cost. In contrast, the ICL approaches often require restoring demonstration examples or incorporating computationally intensive inference algorithms into the base model. As a result, OFA achieves the low inference inference, which is the same as that of zero-shot methods, a key objective for most existing ICL approaches. In addition, we record the practical training and inference cost of Llama3-8B-Instruct on an NVIDIA RTX A6000 for further illustration.

## 5    Conclusion

In this work, we address the problem of few-shot adaptation in Large Language Models (LLMs). We build on the perspective that the forward pass of an LLM can be viewed as an optimization process, and extend this interpretation to a sequence of preconditioned gradient descent steps. Based on this view, we propose tuning the layer-wise preconditioning matrices to improve both convergence speed and generalization, using only a few target-task samples. To this end, two theoretically motivated objective terms are introduced. We evaluate our method across multiple LLMs and benchmark datasets, demonstrating that adaptation with our objective yields substantial performance gains over strong baselines. Our approach also points to a promising direction for low-cost LLM adaptation, particularly in settings with limited data and computational resources.

## Acknowledgements

This work is funded by the EPSRC (grant EP/W031744/1); the InnoHK Hong Kong Centre for Cerebrocardiovascular Engineering (COCHE). DAC was funded by an NIHR Research Professorship; a Royal Academy of Engineering Research Chair; and the InnoHK Hong Kong Centre for Cerebro-cardiovascular Engineering (COCHE); and was supported by the National Institute for Health Research (NIHR) Oxford Biomedical Research Centre (BRC) and the Pandemic Sciences Institute at the University of Oxford.

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

# A  Few-shot performance

We report the entire few-shot performance of all the models, Llama2-7B, Llama3-8B-Instruct, Llama3-8B, and GPT2-XL, in Table 5 to comprehensively evaluate the effectiveness of OFA.

Table 5: Comparison between OFA and other baseline algorithms on LLama2-7B, LLama3-8B-Instruct, LLama3-8B, and GPT2-XL. Mean accuracy and standard deviation across five random seeds are reported. AGnews and DBPedia are not evaluated for GPT2-XL due to its limitation of context window size. **Best** results are highlighted in bold.

| Dataset | SST-2 | SST-5 | TREC | AGNews | Subj | HateSp18 | DBPedia | EmoC | MR |
|---|---|---|---|---|---|---|---|---|---|
| Method | | | | | Llama2-7B | | | | |
| Zero-shot | 83.00 | 27.00 | 50.00 | 70.20 | 51.40 | 54.20 | 72.00 | 41.80 | 73.60 |
| Few-shot (ICL) | $94.44_{\pm1.44}$ | $41.72_{\pm3.68}$ | $77.32_{\pm4.41}$ | $85.68_{\pm2.00}$ | $52.56_{\pm3.09}$ | $70.24_{\pm5.80}$ | $96.64_{\pm0.48}$ | $75.48_{\pm1.63}$ | $93.24_{\pm0.50}$ |
| Soft-prompt | $56.24_{\pm6.99}$ | $24.24_{\pm2.96}$ | $55.20_{\pm4.14}$ | $78.00_{\pm7.60}$ | $57.40_{\pm4.93}$ | $59.56_{\pm6.96}$ | $74.40_{\pm6.43}$ | $35.08_{\pm5.29}$ | $54.32_{\pm1.76}$ |
| Label-anchor | $83.32_{\pm5.95}$ | $27.68_{\pm4.21}$ | $77.48_{\pm3.49}$ | $83.72_{\pm1.04}$ | $53.00_{\pm2.95}$ | $64.52_{\pm8.09}$ | $81.40_{\pm3.67}$ | $59.12_{\pm10.60}$ | $84.40_{\pm5.89}$ |
| Task-vector | $81.44_{\pm4.73}$ | $25.96_{\pm0.59}$ | $65.68_{\pm1.93}$ | $79.68_{\pm4.07}$ | $58.56_{\pm4.91}$ | $67.68_{\pm3.70}$ | $89.48_{\pm2.58}$ | $44.64_{\pm3.53}$ | $82.32_{\pm5.37}$ |
| IA3 | $93.28_{\pm2.29}$ | $46.08_{\pm2.11}$ | $84.40_{\pm5.99}$ | $87.04_{\pm1.97}$ | $71.92_{\pm8.08}$ | $72.44_{\pm2.59}$ | $94.68_{\pm1.09}$ | $64.32_{\pm1.95}$ | $88.80_{\pm2.28}$ |
| I2CL | $87.68_{\pm2.47}$ | $39.12_{\pm2.69}$ | $78.56_{\pm5.32}$ | $85.48_{\pm1.16}$ | $73.84_{\pm3.84}$ | $69.88_{\pm5.67}$ | $90.16_{\pm1.86}$ | $63.72_{\pm1.37}$ | $87.68_{\pm2.26}$ |
| **OFA (Ours)** | $\mathbf{95.84}_{\pm0.41}$ | $\mathbf{50.36}_{\pm3.28}$ | $\mathbf{85.92}_{\pm1.90}$ | $\mathbf{89.00}_{\pm1.26}$ | $\mathbf{88.40}_{\pm4.76}$ | $\mathbf{83.04}_{\pm3.72}$ | $\mathbf{97.72}_{\pm0.52}$ | $\mathbf{76.60}_{\pm2.39}$ | $\mathbf{94.36}_{\pm1.13}$ |
| | | | | | Llama3-8B-Instruct | | | | |
| Zero-shot | 93.00 | 35.80 | 71.00 | 80.40 | 50.80 | 67.80 | 67.40 | 53.60 | 86.40 |
| Few-shot (ICL) | $96.48_{\pm0.48}$ | $46.72_{\pm2.64}$ | $79.92_{\pm5.83}$ | $89.64_{\pm0.59}$ | $57.48_{\pm7.08}$ | $52.72_{\pm2.35}$ | $97.00_{\pm0.28}$ | $65.28_{\pm4.29}$ | $93.12_{\pm0.16}$ |
| Soft-prompt | $84.68_{\pm7.71}$ | $38.40_{\pm5.68}$ | $75.68_{\pm8.17}$ | $84.96_{\pm3.80}$ | $73.28_{\pm5.41}$ | $62.72_{\pm5.54}$ | $87.48_{\pm3.04}$ | $55.32_{\pm9.74}$ | $75.76_{\pm7.71}$ |
| Label-anchor | $93.36_{\pm2.39}$ | $40.54_{\pm5.44}$ | $78.28_{\pm4.07}$ | $84.64_{\pm1.61}$ | $54.16_{\pm2.25}$ | $69.48_{\pm5.43}$ | $87.48_{\pm3.04}$ | $59.36_{\pm2.48}$ | $88.20_{\pm3.69}$ |
| Task-vector | $94.80_{\pm2.02}$ | $56.42_{\pm1.15}$ | $79.83_{\pm1.52}$ | $89.21_{\pm0.58}$ | $76.08_{\pm1.23}$ | $67.12_{\pm0.32}$ | $79.52_{\pm1.84}$ | $57.96_{\pm4.59}$ | $86.52_{\pm0.64}$ |
| IA3 | $94.32_{\pm0.82}$ | $49.24_{\pm2.06}$ | $87.60_{\pm3.46}$ | $88.36_{\pm1.80}$ | $82.04_{\pm7.43}$ | $77.20_{\pm4.37}$ | $92.56_{\pm1.82}$ | $68.04_{\pm2.24}$ | $91.76_{\pm0.43}$ |
| I2CL | $90.84_{\pm0.98}$ | $48.96_{\pm2.48}$ | $79.60_{\pm6.22}$ | $88.96_{\pm2.03}$ | $81.48_{\pm4.68}$ | $65.88_{\pm3.61}$ | $91.20_{\pm2.03}$ | $64.32_{\pm2.05}$ | $88.88_{\pm0.61}$ |
| **OFA (Ours)** | $\mathbf{97.08}_{\pm0.27}$ | $\mathbf{58.32}_{\pm2.74}$ | $\mathbf{89.06}_{\pm1.49}$ | $\mathbf{91.84}_{\pm0.61}$ | $\mathbf{92.64}_{\pm3.43}$ | $\mathbf{89.47}_{\pm0.47}$ | $\mathbf{97.92}_{\pm1.06}$ | $\mathbf{79.24}_{\pm4.87}$ | $\mathbf{94.56}_{\pm0.51}$ |
| Method | | | | | Llama3-8B | | | | |
| Zero-shot | 56.00 | 33.20 | 66.40 | 85.80 | 50.60 | 50.80 | 55.80 | 40.60 | 53.80 |
| Few-shot (ICL) | $95.32_{\pm0.74}$ | $44.36_{\pm1.93}$ | $74.48_{\pm6.17}$ | $87.20_{\pm1.04}$ | $63.84_{\pm8.27}$ | $70.60_{\pm5.92}$ | $85.56_{\pm3.67}$ | $52.30_{\pm3.62}$ | $91.88_{\pm0.86}$ |
| Soft-prompt | $59.44_{\pm12.5}$ | $28.44_{\pm6.93}$ | $70.32_{\pm10.62}$ | $85.68_{\pm2.58}$ | $69.12_{\pm9.85}$ | $63.20_{\pm4.88}$ | $85.36_{\pm6.45}$ | $54.20_{\pm11.79}$ | $60.28_{\pm11.59}$ |
| Label-anchor | $84.14_{\pm0.20}$ | $35.44_{\pm0.48}$ | $77.68_{\pm2.90}$ | $86.20_{\pm1.81}$ | $64.40_{\pm0.38}$ | $68.08_{\pm1.27}$ | $74.24_{\pm2.71}$ | $59.72_{\pm3.64}$ | $84.28_{\pm0.97}$ |
| Task-vector | $94.28_{\pm8.96}$ | $37.20_{\pm2.83}$ | $75.80_{\pm1.50}$ | $85.00_{\pm3.74}$ | $68.40_{\pm0.80}$ | $55.60_{\pm3.41}$ | $73.28_{\pm1.27}$ | $54.64_{\pm0.99}$ | $75.28_{\pm4.70}$ |
| IA3 | $92.72_{\pm1.58}$ | $46.40_{\pm2.60}$ | $80.04_{\pm2.85}$ | $85.44_{\pm2.63}$ | $69.24_{\pm6.15}$ | $62.64_{\pm3.86}$ | $83.20_{\pm3.93}$ | $64.36_{\pm3.16}$ | $89.52_{\pm1.48}$ |
| I2CL | $87.36_{\pm3.21}$ | $39.32_{\pm4.02}$ | $77.72_{\pm6.99}$ | $85.20_{\pm2.32}$ | $70.03_{\pm5.39}$ | $58.08_{\pm9.79}$ | $86.44_{\pm2.41}$ | $62.64_{\pm5.96}$ | $86.84_{\pm7.29}$ |
| **OFA (Ours)** | $\mathbf{96.92}_{\pm0.35}$ | $\mathbf{54.96}_{\pm3.29}$ | $\mathbf{87.52}_{\pm4.40}$ | $\mathbf{90.36}_{\pm0.93}$ | $\mathbf{91.44}_{\pm2.34}$ | $\mathbf{86.76}_{\pm5.71}$ | $\mathbf{97.76}_{\pm0.45}$ | $\mathbf{78.86}_{\pm5.85}$ | $\mathbf{94.04}_{\pm0.34}$ |
| Method | | | | | GPT2-XL | | | | |
| Zero-shot | 74.76 | 30.44 | 35.40 | – | 64.88 | 70.84 | – | 37.88 | 71.36 |
| Few-shot (ICL) | $73.65_{\pm8.89}$ | $35.95_{\pm2.39}$ | $60.64_{\pm5.00}$ | – | $63.82_{\pm10.55}$ | $51.86_{\pm3.22}$ | – | $38.62_{\pm6.87}$ | $75.79_{\pm9.25}$ |
| Soft-prompt | $61.04_{\pm3.45}$ | $23.96_{\pm2.09}$ | $40.60_{\pm10.15}$ | – | $55.44_{\pm4.12}$ | $63.92_{\pm7.06}$ | – | $36.68_{\pm2.70}$ | $57.60_{\pm3.53}$ |
| Label-anchor | $63.40_{\pm8.82}$ | $22.36_{\pm3.37}$ | $66.36_{\pm10.69}$ | – | $55.56_{\pm4.26}$ | $54.88_{\pm4.53}$ | – | $36.68_{\pm2.70}$ | $60.20_{\pm3.32}$ |
| Task-vector | $81.08_{\pm4.87}$ | $28.52_{\pm1.37}$ | $41.40_{\pm5.35}$ | – | $71.81_{\pm1.86}$ | $62.48_{\pm2.83}$ | – | $37.60_{\pm2.48}$ | $78.40_{\pm2.26}$ |
| IA3 | $86.64_{\pm2.89}$ | $40.52_{\pm2.25}$ | $70.96_{\pm8.61}$ | – | $71.52_{\pm8.46}$ | $70.84_{\pm3.63}$ | – | $62.24_{\pm3.50}$ | $83.24_{\pm1.09}$ |
| I2CL | $80.16_{\pm3.98}$ | $35.04_{\pm2.60}$ | $51.48_{\pm5.26}$ | – | $65.96_{\pm4.83}$ | $68.32_{\pm4.76}$ | – | $47.92_{\pm1.84}$ | $83.20_{\pm3.29}$ |
| **OFA (Ours)** | $\mathbf{88.68}_{\pm2.66}$ | $\mathbf{42.48}_{\pm2.51}$ | $\mathbf{70.60}_{\pm6.44}$ | – | $\mathbf{86.11}_{\pm4.29}$ | $\mathbf{71.44}_{\pm8.65}$ | – | $\mathbf{65.30}_{\pm4.18}$ | $\mathbf{84.80}_{\pm6.21}$ |

# B  Proof of Theorem 3.1

**Theorem 3.1.** *Let $f : \mathbb{R}^d \to \mathbb{R}$ be a twice continuously differentiable function with locally Lipschitz gradients. Suppose the update rule is given by:*

$$Z_{t+1} = Z_t - P_t \nabla \mathcal{L}(Z_t),$$

*where each $P_t \in \mathbb{R}^d \times \mathbb{R}^d$ is a learnable preconditioning matrix. Define the step-ratio objective in Eq. 3 Under the assumption that $f$ admits a local second-order Taylor expansion approximation at each step, then minimizing $\mathcal{J}(P)$ encourages the learned preconditioners $P_t$ to induce local operators $I - \eta P_t H_t$ with $H_t = \nabla^2 f(Z_t)$ with a smaller spectral radius.*

$$\|Z_{t+1} - Z^*\| \leq \rho_t \|Z_t - Z^*\|, \ \rho_t < \rho_{t-1}.$$

*Thus, it induces faster local contraction and improved convergence.*

*Proof.* By Taylor's theorem, for a smooth function f, near point $x_t$, we have:

$$f(x) = f(x_t) + \nabla f(x_t)^T (x - x_t) + \frac{1}{2}(x - x_t)^T H_t (x - x_t).$$

Given the preconditioned gradient descent:

$$x_{t+1} - x_t = -\eta P_t \nabla f(x_t),$$

with the quadratic approximation, we approximate the gradient:

$$\nabla f(x_t) \approx H_t(x_t - x^*),$$

then

$$x_{t+1} - x_t = -\eta P_t H_t(x_t - x^*),$$

and

$$x_{t+1} - x^* = (I - \eta P_t H_t)(x_t - x^*).$$

Then the step-ratio objective becomes:

$$\mathcal{J}(\theta) = \sum_{t=1}^{T-1} \frac{\|x_t - x_{t+1}\|}{\|x_t - x_{t-1}\|} = \sum_{t=1}^{T-1} \frac{\| - \eta P_t H_t(x_t - x^*)\|}{\|x_t - x_{t-1}\|},$$

and operator $I - \eta P_t H_t$ governs convergence. Assuming:

$$\rho_t = \text{spectral radius}(I - \eta P_t H_t) < 1.$$

Then minimizing $\mathcal{J}(P))$ ensures $\rho_t$ decreases over time:

$$x_{t+1} - x^* = (I - \eta P_t H_t)(x_t - x^*),$$

which leads to

$$\|x_{t+1} - x^*\| = \|(I - \eta P_t H_t)(x_t - x^*)\| \leq \rho_t \|x_t - x^*\|, \ \rho_t < \rho_{t-1}.$$

$\square$

## C   Proof of Theorem

**Theorem 3.2.** *Let $Z_T$ be the final parameters after $T$ steps of optimization, with preconditioning update rules in Eq. 2 and denoting $\nabla^2 \mathcal{L}_{train}(Z_t)$ as the Hessian at step $t$ with $\|P_t \nabla^2 \mathcal{L}_{train}(Z_t)\|_F$ measuring the curvature after preconditioning at that step. Assume the loss is smooth, $\|\nabla^2 \mathcal{L}(Z_t)\|_F \leq \mu$, and the gradient is bounded, $\|\nabla \mathcal{L}(Z_t)\| \leq G$, the generalization gap satisfies:*

$$\mathbb{E}[\mathcal{L}_{test}(Z_T) - \mathcal{L}_{train}(Z_T)] \leq \mathcal{O}\left(\sqrt{\frac{1}{n}\sum_{t=1}^{T} \|P_t \nabla^2 \mathcal{L}_{train}(Z_t)\|_F^2}\right).$$

*Proof.* The proof follows from stability-based generalization bounds and Taylor expansion.

Let $\Delta_t = \theta_{t+1} - \theta_t = -\eta P_t \nabla \mathcal{L}_{train}(\theta_t)$.

By a second-order Taylor approximation, for a perturbation $\epsilon$:

$$\mathcal{L}(Z + \epsilon) \approx \mathcal{L}(Z) + \nabla \mathcal{L}(Z)^T \epsilon + \frac{1}{2}\epsilon^T \nabla^2 \mathcal{L}(Z)\epsilon.$$

Consider the increase in loss under Gaussian perturbation $\epsilon \sim \mathcal{N}(0, \Sigma)$, used in PAC-Bayes analysis. The expected curvature-based increase is:

$$\mathbb{E}[\mathcal{L}(Z_T + \epsilon) - \mathcal{L}(Z_T)] \approx \frac{1}{2}\text{Tr}(\Sigma \nabla^2 \mathcal{L}(Z_T)).$$

Since $\Sigma$ is shaped by optimization history through $\{\Delta_t\}_{t=1}^{T}$ [17, 47]. Then, the effective curvature encountered is influenced by the preconditioned curvature norm:

$$\|\Delta_t^T \nabla^2 \mathcal{L}_{train}(\theta_t)\Delta_t\| = \eta^2 \nabla \mathcal{L}(\theta_t)^T P_t \nabla^2 \mathcal{L}_{train}(\theta_t) P_t \nabla \mathcal{L}(\theta_t) \leq \eta^2 G^2 \|P_t \nabla^2 \mathcal{L}_{train}(\theta_t)\|_F^2.$$

Summing over $t = 1$ to $T$, we obtain:

$$\sum_{t=1}^{T} \|\Delta_t^T \nabla^2 \mathcal{L}_{\text{train}}(Z_t)\Delta_t\| \leq \eta^2 G^2 \sum_{t=1}^{T} \|P_t \nabla^2 \mathcal{L}_{\text{train}}(Z_t)\|_F^2.$$

Applying a Rademacher complexity or PAC-Bayes-based argument [19], this leads to:

$$\mathbb{E}[\mathcal{L}_{\text{test}}(Z_T) - \mathcal{L}_{\text{train}}(Z_T)] \leq \mathcal{O}\left(\sqrt{\frac{1}{n}\sum_{t=1}^{T} \|P_t \nabla^2 \mathcal{L}_{\text{train}}(Z_t)\|_F^2}\right).$$

$\square$

## D    Prompting Templates

Table 6: Illustration of prompting templates and label spaces in our setting. The input prompt template is decomposed into multiple {Sentence} and {Label} pairs, which are placeholders for the input sentence and its corresponding label. The template containing a single example for each dataset is generated for the illustration, while in the multiple demonstration example setting, the sentence-label pairs are stacked and separated by a newline character: '\n'.

| Dataset | Template | Label Space |
|---------|----------|-------------|
| SST-2 | Review: {Sentence}
Sentiment: {Label} | negative / positive |
| SST-5 | Sentence: {Sentence}
Sentiment: {Label} | terrible / negative / neutral / positive / great |
| MR | Review: {Sentence}
Sentiment: {Label} | negative / positive |
| Subj | Sentence: {Sentence}
Label: {Label} | objective / subjective |
| DBPedia | Input: {Sentence}
Label: {Label} | company / school / artist / athlete / politics / transportation / building / nature / village / animal / plant / album / film / book |
| AGNews | News: {Sentence}
Type: {Label} | World / Sports / Business / Technology |
| TREC | Question: {Sentence}
Answer Type: {Label} | Abbreviation / Entity / Person / Location / Number |
| HateSpeech18 | Text: {Sentence}
Label: {Label} | neutral / hate |
| EmoC | Dialogue: {Sentence}
Emotion: {Label} | others / happy / sad / angry |

**Extra Details** We follow the dataset preprocessing protocol from [39] for our experiments setting. Regarding HateSpeech18, only the first two categories—{0: neutral} and {1: hate} are used, since the very few number of samples in the other two may impede a comprehensive evaluation of the model in the test stage.

## E    LoRA experiment settings

We describe the details of the LoRA implementation in our experiments. For a fair comparison, the LoRA model trained for each individual dataset is tuned by grid search according to the hyperparameter pool, including LoRA alpha, LoRA dropout, optimizer, and learning rate in Table 7.

Table 7: Hyperparameter Pool for the LoRA model tuning.

| Hyperparameter | Values |
| --- | --- |
| LoRA alpha | 8, 16, 32, 64 |
| LoRA dropout | 0.0, 0.05, 0.1 |
| Optimizer | AdamW |
| Learning rate | 0.001, 0.0001, 0.00001 |

# F  Hyperparameter Pool

We conduct the grid search for fair comparison over all the models, including all the baseline models and ours. The hyperparameter pool for the model tuning is give in Table 8.

Table 8: Hyperparameter Pool for the LoRA model tuning.

| Hyperparameter | Values |
| --- | --- |
| $\lambda_1$ | 0.1, 0.001, 0.0001, 0.00001, 0.000001 |
| $\lambda_2$ | 0.1, 0.001, 0.0001, 0.00001, 0.000001 |
| Optimizer | AdamW |
| Learning rate | 0.001, 0.0001, 0.00001, |
| Weight decay | 0.001, 0.0001, 0.00001, 0.000001 |
| Training epoch | 20, 50, 60, 80, 100 |

# G  Limitation and Future Work

In this work, we address the problem of few-shot adaptation within the LLM framework by enhancing both the internal optimization efficiency and the generalization capability of pretrained models. Specifically, we introduce two distinct objective terms, each targeting one of these properties. While this design improves performance, it also increases the burden of hyperparameter tuning and computational overhead. We leave the unification of these objectives into a single term, enabling joint optimization of both properties, as future work. Moving forward, we aim to contribute to the community by developing a rigorous theoretical foundation for this adaptation problem and further improving our method based on these insights.

# H  Broader impacts

This paper aims to contribute to the advancement of Machine Learning, especially to the few-shot adaptation of LLMs. While our work may have various societal implications, none require specific emphasis in this context.

