# OpenReview forum: "Optimization Inspired Few-Shot Adaptation for Large Language Models"
_NeurIPS.cc/2025/Conference — NeurIPS 2025 spotlight_

### Official Review · Reviewer_7e2o · 2025-06-30

**Clarity:** 2
**Significance:** 2
**Originality:** 3
**Rating:** 4
**Confidence:** 3

**Summary:**

This paper introduces Optimization-Inspired Few-Shot Adaptation (OFA), an approach for adapting LLMs to new tasks with limited data. The paper reinterpret the forward pass of LLMs as a sequence of preconditioned gradient descent steps and leverage this insight to propose a parameter-efficient adaptation method that modifies only the LayerNorm parameters. OFA combines three objectives: cross-entropy loss, a step-ratio objective that improves optimization efficiency by promoting smoother convergence paths, and a sharpness minimization objective that enhances generalization by steering optimization toward flat minima. The method is theoretically grounded with convergence and generalization bounds, while being extremely parameter-efficient (using only 0.27M parameters). Extensive experiments across multiple datasets and model architectures demonstrate that OFA consistently outperforms existing ICL and PEFT methods, achieving 4-10% improvements over baseline approaches without introducing additional inference overhead.

**Questions:**

- In your experimental evaluation, you use IA3 as one of the baseline methods for comparison. However, the original IA3 paper (citation [35]) introduces additional training components beyond the parameter-efficient architecture, specifically Unlikelihood Training and Length Normalization, which are designed to address challenges in few-shot learning scenarios. Did you implement these additional loss terms when training your IA3 baseline? If not, do you believe including these components might narrow the performance gap between IA3 and your proposed OFA method?
- The paper introduces novel loss terms (step-ratio objective and sharpness minimization) that are theoretically motivated to improve optimization efficiency and generalization. Did you consider applying these loss terms to other parameter-efficient fine-tuning methods such as IA3 or LoRA to isolate their effect?

**Ethical Concerns:**

["NO or VERY MINOR ethics concerns only"]

**Final Justification:**

The authors have addressed my concerns regarding computational cost and have provided additional experiments on adding those loss terms during LoRA training, showing that their method consistently outperforms the baseline methods considered. Accordingly, I have decided to increase my score by one point.

**Limitations:**

yes

**Quality:**

3

**Strengths And Weaknesses:**

### Strengths
- The paper provides an interesting theoretical interpretation of transformer layers as implementing gradient descent steps, creating a good foundation for their method.
- OFA achieves impressive results while updating only LayerNorm parameters (0.27M parameters), making it significantly more efficient than competing approaches.
- The experiments cover multiple datasets and model architectures (Llama2-7B, Llama3-8B, GPT2-XL), demonstrating consistent performance improvements.
- The paper provides theoretical analysis connecting their objectives to convergence speed and generalization bounds.

### Weaknesses
- citation [35] on line 27 doesn't adequately support the claim that PEFT methods suffer from overfitting in few-shot scenarios. This weakens one of their motivating arguments.
- The Hutchinson approximation for sharpness estimation adds implementation complexity that isn't fully addressed in terms of computational overhead during training.
- Some of the theoretical results rely on assumptions like locally Lipschitz gradients and quadratic approximations that may not always hold in practice for complex LLMs.

---

> ### Author Rebuttal · Authors · 2025-07-30
>
> > Weakness 1: Not sufficient support for the claim that the PEFT models, such as LoRA, suffer from overfitting in few-shot scenarios.
>
> We appreciate the reviewer for pointing out this issue. Actually, there are more studies that have pointed out the overfitting problem of LoRA in few-shot settings.
>
> Specifically, [1] notes that “the matrix decomposition approximation might impact the adaptability” of LoRA, as its low-rank constraint can hinder generalization when data is extremely scarce. To address this, [2] proposes HyperLoRA, a meta-adaptation method designed to overcome the challenges of “few-shot adaptation” with static LoRA modules. Similarly, [3] introduces BiLoRA, a bi-level optimization framework motivated by the observation that standard LoRA “often overfit training data, resulting in sub-optimal generalization on test data.” Furthermore, several studies [4–9] conclude that LoRA alone is insufficient for generalization when the adaptation data is scarce. This highlights LoRA’s limited adaptability when learning from very small datasets. We also provide practical evidence in our response to Question 2 to support this argument.
>
> > Weakness 2: Computational cost from our sharpness approximation.
>
> We acknowledge that estimating layer-wise sharpness incurs additional computational overhead. However, this cost is confined to the training phase and remains acceptable when compared to baseline models. At inference time, our method maintains the same efficiency as standard zero-shot settings, offering significantly lower complexity than many baselines.
> We compare the training costs of various algorithms on DBPedia, the dataset with the largest number of categories in our experimental suite, using LLaMA3-8B-Instruct.
>
> Compared to LoRA-based models, our method achieves faster training times and better performance, while maintaining the same inference cost. When compared with embedding-based approaches such as Label-Anchor, Task-Vector, and I2CL, our method incurs slightly higher training time but achieves significantly better performance during inference, along with reduced inference time. Finally, in comparison to IA3 and Soft-Prompt, our method demonstrates substantial improvements in both inference efficiency and accuracy.
>
> | Method           | Soft-prompt | Label-anchor | Task-vector | IA3     | I2CL    | LoRA (Rank16) | LoRA (Rank1) | Ours    |
> |------------------|-------------|---------------|--------------|---------|---------|---------------|----------------|---------|
> | Training cost (s) | 196.08      | 50.41         | 213.25       | 324.12  | 312.50  | 483.52       |  449.23       | 367.05  |
> | Inference cost (s) | 53.64       | 52.41         | 56.78        | 52.40   | 52.59   | 51.38         | 51.35          | 51.37   |
> |Performance (Accuracy %)| 82.88 | 87.48| 79.52 | 92.56 | 91.20 | 92.62 | 57.88  | 97.92 |
>
>
> > Weakness 3: The assumption about locally Lipschitz gradients and quadratic approximations for analysis.
>
> Our local Lipschitz gradient assumption is supported by recent theoretical work [10], which shows that the attention mechanism is smooth under standard conditions. In our framework, the gradient is implicitly computed as each attention layer corresponds to a single gradient step in the internal optimization process.
>
> Similarly, our assumption of second-order smoothness in in-context learning is theoretically supported by [11,12]. In practice, computing gradients by backpropagation with respect to the input of each transformer layer equals the second-order differentiability of the loss with respect to internal representations, thereby justifying our use of a second-order Taylor expansion for approximation. Moreover, since the internal optimization is implicitly performed by the attention layers, the backpropagation used to update LLM parameters effectively captures second-order information. Besides the in-context learning setting, the second-order assumption for conventional neural network settings is also broadly studied and applied [13, 14], which can be seen as third order in our case.
>
> We will include more discussion about the validity of the assumptions in the revised version.
>
> > Question 1: Do we follow the implementation of IA3?
>
> We implement IA3 following its default architectural design and training objective, including unlikelihood training and length normalization terms, to ensure a fair comparison. Nevertheless, our model consistently demonstrates a clear performance advantage across all experiments.
>
> > Question 2: Can our loss be applied to other algorithms, such as LoRA?
>
> We appreciate this question. Regarding the performance of combining LoRA with our proposed loss in the few-shot learning setting, please refer to the tables below as well as Table 2 in our submission, which includes additional results across different LoRA ranks. From these results, we observe that LoRA generally underperforms in the few-shot setting due to limited data availability for effectively fitting the model. Compared to LoRA trained with standard cross-entropy loss, our loss helps mitigate this overfitting to some extent, though it does not fully resolve the problem.
>
>  The results from the table below show that the LoRA models suffer in the few-shot settings.
>
> ### LoRA comparsion on Llama2-7B
>
> | Method  | SST-2| SST-5| TREC| AGNews  | Subj | HateSpeech18| DBPedia| EmoC| MR |
> |-------|------|-----|-----|-------|-------|-----|-----|----|----|
> | Rank 1  | $89.64_{\pm 3.23}$ | $24.84_{\pm 9.92}$ | $22.88_{\pm 9.09}$ | $50.56_{\pm 31.36}$ | $72.08_{\pm 8.74}$ | $69.14_{\pm 9.76}$ | $59.44_{\pm 42.01}$ | $25.64_{\pm 2.95}$ | $64.84_{\pm 13.86}$ |
> | Rank1 (our loss)    | $88.48_{\pm 3.34}$ | $20.80_{\pm 0.89}$ | $24.32_{\pm 6.02}$ | $62.04_{\pm 29.5}$  | $72.84_{\pm 11.33}$ | $69.38_{\pm 10.77}$ | $74.6_{\pm 33.23}$  | $33.24_{\pm 18.28}$ | $64.88_{\pm 18.48}$ |
> | Ours | $\mathbf{95.84}_{\pm0.41}$ | $\mathbf{50.36}_{\pm3.28}$ | $\mathbf{85.92}_{\pm1.90}$ | $\mathbf{89.00}_{\pm1.26}$ | $\mathbf{88.40}_{\pm4.76}$ | $\mathbf{83.04}_{\pm3.72}$ | $\mathbf{97.72}_{\pm0.52}$ | $\mathbf{76.60}_{\pm2.39}$ | $\mathbf{94.36}_{\pm1.13}$ |
>
> ### LoRA comparison on Llama3-8B-Instruct
>
> | Method  | SST-2 | SST-5| TREC | AGNews | Subj | HateSpeech18 | DBPedia | EmoC | MR |
> |-------|------|-----|--------|-----|-----|------|-----|-----|------|
> | Rank 1 | $87.08_{\pm 4.81}$ | $19.52_{\pm 0.45}$ | $25.88_{\pm 9.17}$ | $39.72_{\pm 25.1}$ | $78.92_{\pm 8.43}$ | $63.60_{\pm 9.07}$ | $57.88_{\pm 39.37}$ | $24.68_{\pm 1.17}$ | $87.20_{\pm 3.38}$ |
> | Rank1 (our loss)    | $87.40_{\pm 8.05}$ | $20.32_{\pm 1.72}$ | $27.52_{\pm 6.09}$ | $39.12_{\pm 24.40}$ | $80.96_{\pm 5.55}$ | $68.92_{\pm 8.33}$ | $73.52_{\pm 32.69}$ | $27.68_{\pm 3.65}$ | $87.84_{\pm 13.59}$ |
> | Ours  | $\mathbf{97.08}_{\pm0.27}$ | $\mathbf{58.32}_{\pm2.74}$ | $\mathbf{89.06}_{\pm1.49}$ | $\mathbf{91.84}_{\pm0.61}$ | $\mathbf{92.64}_{\pm3.43}$ | $\mathbf{89.47}_{\pm0.47}$ | $\mathbf{97.92}_{\pm1.06}$ | $\mathbf{79.24}_{\pm4.87}$ | $\mathbf{94.56}_{\pm0.51}$ |
>
> We hypothesize three main reasons for this behavior: 1) Parameter count: Even in the rank-1 setting, LoRA introduces twice as many parameters as our method, thus requiring more data to generalize effectively. 2) Initialization: Our parameters are pretrained on a large-scale dataset, whereas LoRA parameters are initialized from scratch, making our method more data-efficient for adaptation. 3) Optimization behavior: LoRA significantly alters the gradient flow, the core component of the optimization process, whereas our method preserves the original gradients and instead adapts the preconditioning matrices to each task.
>
> [1] Srinivasan, Krishna Prasad Varadarajan, et al. "Comparative analysis of different efficient fine tuning methods of large language models (llms) in low-resource setting." arXiv preprint arXiv:2405.13181.2024..
>
> [2]Lv, Chuancheng, et al. "Hyperlora: Efficient cross-task generalization via constrained low-rank adapters generation." Findings: EMNLP 2024.
>
> [3] Qiang, Rushi, Ruiyi Zhang, and Pengtao Xie. "Bilora: A bi-level optimization framework for overfitting-resilient low-rank adaptation of large pre-trained models." arXiv preprint arXiv:2403.13037. 2024.
>
> [4]Kong, Xiaoyu, et al. "Customizing language models with instance-wise lora for sequential recommendation." NeurIPS. 2024.
>
> [5] Asadi, Nader, et al. "Does combining parameter-efficient modules improve few-shot transfer accuracy?." arXiv preprint arXiv:2402.15414. 2024.
>
> [6] Baklouti, Ghassen, et al. "Regularized Low-Rank Adaptation for Few-Shot Organ Segmentation." arXiv preprint arXiv:2507.15793. 2025.
>
> [7] Asadi, Nader, et al. "Combining Pre-trained LoRA Modules Improves Few-shot Adaptation of Foundation Models to New Tasks." ICML 2024 Workshop on Foundation Models in the Wild.
>
> [8]Farina, Matteo, et al. "Rethinking Few-Shot Adaptation of Vision-Language Models in Two Stages." CVPR. 2025.
>
> [9] Li, Yang, Shaobo Han, and Shihao Ji. "Vb-lora: Extreme parameter efficient fine-tuning with vector banks." NeurIPS. 2024.
>
> [10] Castin, Valérie, Pierre Ablin, and Gabriel Peyré. "How Smooth Is Attention?" ICML. 2024.
>
> [11] Fu, Deqing, et al. "Transformers learn to achieve second-order convergence rates for in-context linear regression." NeurIPS. 2024.
>
> [12] Giannou, Angeliki, et al. "How Well Can Transformers Emulate In-context Newton's Method?." arXiv preprint arXiv:2403.03183. 2024.
>
> [13] Tatzel, Lukas, et al. "Debiasing Mini-Batch Quadratics for Applications in Deep Learning." ICLR. 2025.
>
> [14] Petersen, Felix, et al. "Newton losses: Using curvature information for learning with differentiable algorithms." NeurIPS. 2024.
>
> [15] Shin, Dahun, et al. "Sassha: Sharpness-aware Adaptive Second-order Optimization with Stable Hessian Approximation." ICML. 2025.

---

> > ### Author Response · Authors · 2025-08-04
> >
> > Dear Reviewer,
> >
> > We appreciate the time and effort you have taken to review our submission and provide valuable feedback. We have carefully addressed the comments raised during the first round and submitted our responses accordingly. As the discussion period is ending soon, we would greatly appreciate any further feedback you might have. Please let us know if there are any remaining concerns or questions, so we can address them in time.
> >
> > Thank you again for reviewing our paper.
> >
> > Best regards,
> >
> > Authors

---

> > > ### Comment · Reviewer_7e2o · 2025-08-05
> > >
> > > Thank you for addressing my questions and acknowledging the identified weaknesses. I appreciate the clarifications and experimental results provided by the authors, which have addressed my concerns. As a result, I am increasing my score by one point.

---

> > > > ### Author Response · Authors · 2025-08-05
> > > >
> > > > Thank you very much for your support and feedback, and thank you for increasing your score. We are glad to know that we have addressed your comments.

---

### Official Review · Reviewer_CPuh · 2025-07-01

**Clarity:** 3
**Significance:** 3
**Originality:** 3
**Rating:** 4
**Confidence:** 2

**Summary:**

The paper proposes a new few-shot adaptation algorithm that adjusts the parameters of layer-norm on few-shot data. This approach is motivated by the gradient descent properties of transformer layers to encourage smooth gradient behaviors across layers (step ratio regularization). The proposed approach also aims to learn flat minima using the Hutchinson approximation method. Empirical results show the proposed approach outperforms other baselines: ICL, soft prompt, label anchor, task vector, IA3, I2CL, and LoRA. In particular, it outperforms LoRA by a large margin while using less parameters. Ablation study through probe classifier analysis shows the importance of step ratio and sharpness regularization.

**Questions:**

1. Are the proposed terms applicable to any SFT or PEFT training algorithm?

2. How many exemplars are used in the test sets? How the proposed approach would work if we increase the number of exemplars, and at what point it would become limited by the representation of the layer-norm parameters.

3. How to set the hyperparameters of different loss terms and is the model sensitive to these parameters?

**Ethical Concerns:**

["NO or VERY MINOR ethics concerns only"]

**Final Justification:**

My final rating is borderline accept. The paper presents clear motivation and shows strong empirical results. The authors' response addressed my concerns regarding the distinctions between soft prompting and LoRA. The additional experiments exploring sample size variations provide valuable insights for practitioners weighing different approaches based on the level of available training data.

**Limitations:**

yes

**Quality:**

3

**Strengths And Weaknesses:**

Strength

1. The paper is well-motivated. The proposed regularization terms on step ratio and loss sharpness are interesting and technically sound.
2. The proposed approach demonstrates strong empirical results, outperforming LoRA by a large margin while using less parameters.
3. The paper carries out comprehensive ablation studies and visualization to demonstrate the value of step ratio and sharpness regularization.

Weakness

1. Loose connection between regularization and parameter efficient fine-tuning. The proposed approach is comprised of cross entropy loss on the layer norm parameters, and two regularization terms. It seems these regularization terms could be applied just as easily to other PEFT (or any training) methods, such as soft prompt or LoRA.
2. Scaling with exemplar size. It is unclear how the proposed approach would work if we increase the number of exemplars, and at what point it would become limited by the representation of the layer-norm parameters.

---

> ### Author Rebuttal · Authors · 2025-07-29
>
> > Weakness 1: Loose connection between the introduced loss terms and the efficient fine-tuning. Application of our loss to Soft Prompt and LoRA.
>
> Admittedly, soft prompt, LoRA, and our proposed regularization terms are performed in different aspects. However, LoRA and soft prompts have drawbacks in few-shot adaptations. LoRA adapter introduces a substantial number of additional parameters, making it less suitable for few-shot learning scenarios, a limitation that our loss function does not inherently resolve. The in-context learning based algorithms usually have limited flexibility, such as suitable parameters, for adaptation to new tasks.
>
> In our work, we primarily focus on the few-shot learning setting, where the training data is too limited to support effective supervised fine-tuning (SFT) or parameter-efficient fine-tuning (PEFT) methods such as LoRA. Our experimental results show that LoRA often fails to outperform even the zero-shot and few-shot baselines. This is because the LoRA adapter introduces a substantial number of additional parameters, making it less suitable for few-shot learning scenarios, a limitation that our loss function does not inherently resolve. Our experimental results support this assessment: although LoRA models trained with our loss consistently outperform those trained with the LoRA only models, they still fall short of outperforming ours. This pattern is observed consistently across both the LLaMA-2 and LLaMA-3 model families.
>
> ### LoRA comparsion on Llama2-7B. We quote the results from Table 2 in our original submission.
>
> | Method              | SST-2             | SST-5             | TREC              | AGNews            | Subj              | HateSpeech18      | DBPedia           | EmoC              | MR                |
> |---------------------|-------------------|-------------------|-------------------|-------------------|-------------------|-------------------|-------------------|-------------------|-------------------|
> | Rank 1              | $89.64_{\pm 3.23}$ | $24.84_{\pm 9.92}$ | $22.88_{\pm 9.09}$ | $50.56_{\pm 31.36}$ | $72.08_{\pm 8.74}$ | $69.14_{\pm 9.76}$ | $59.44_{\pm 42.01}$ | $25.64_{\pm 2.95}$ | $64.84_{\pm 13.86}$ |
> | Rank1 (our loss)    | $88.48_{\pm 3.34}$ | $20.80_{\pm 0.89}$ | $24.32_{\pm 6.02}$ | $62.04_{\pm 29.5}$  | $72.84_{\pm 11.33}$ | $69.38_{\pm 10.77}$ | $74.6_{\pm 33.23}$  | $33.24_{\pm 18.28}$ | $64.88_{\pm 18.48}$ |
> | Ours                | $\mathbf{95.84}_{\pm0.41}$ | $\mathbf{50.36}_{\pm3.28}$ | $\mathbf{85.92}_{\pm1.90}$ | $\mathbf{89.00}_{\pm1.26}$ | $\mathbf{88.40}_{\pm4.76}$ | $\mathbf{83.04}_{\pm3.72}$ | $\mathbf{97.72}_{\pm0.52}$ | $\mathbf{76.60}_{\pm2.39}$ | $\mathbf{94.36}_{\pm1.13}$ |
>
>
> ### LoRA comparison on Llama3-8B-Instruct. We quote the results from Table 2 in our original submission.
>
> | Method              | SST-2             | SST-5             | TREC              | AGNews            | Subj              | HateSpeech18      | DBPedia           | EmoC              | MR                |
> |---------------------|-------------------|-------------------|-------------------|-------------------|-------------------|-------------------|-------------------|-------------------|-------------------|
> | Rank 1              | $87.08_{\pm 4.81}$ | $19.52_{\pm 0.45}$ | $25.88_{\pm 9.17}$ | $39.72_{\pm 25.1}$ | $78.92_{\pm 8.43}$ | $63.60_{\pm 9.07}$ | $57.88_{\pm 39.37}$ | $24.68_{\pm 1.17}$ | $87.20_{\pm 3.38}$ |
> | Rank1 (our loss)    | $87.40_{\pm 8.05}$ | $20.32_{\pm 1.72}$ | $27.52_{\pm 6.09}$ | $39.12_{\pm 24.40}$ | $80.96_{\pm 5.55}$ | $68.92_{\pm 8.33}$ | $73.52_{\pm 32.69}$ | $27.68_{\pm 3.65}$ | $87.84_{\pm 13.59}$ |
> | Ours                | $\mathbf{97.08}_{\pm0.27}$ | $\mathbf{58.32}_{\pm2.74}$ | $\mathbf{89.06}_{\pm1.49}$ | $\mathbf{91.84}_{\pm0.61}$ | $\mathbf{92.64}_{\pm3.43}$ | $\mathbf{89.47}_{\pm0.47}$ | $\mathbf{97.92}_{\pm1.06}$ | $\mathbf{79.24}_{\pm4.87}$ | $\mathbf{94.56}_{\pm0.51}$ |
>
> From an optimization perspective, Soft Prompt modifies the initialization state, whereas our loss function influences the optimization trajectory itself by adapting the preconditioning matrices. To this end, the gradient generated by our loss would not be informative for the target learnable parameters. However, we still apply our loss to the soft prompt. Therefore, the soft prompt problem cannot be solved by our algorithm, and in the empirical results, it harms the performance of our model.
>
> | Method                     | SST‑2  | SST‑5 | TREC  | AGNews | Subj  | HateSp18 | DBPedia | EmoC  | MR    |
> |---------------------------|--------|-------|-------|--------|-------|-----------|---------|-------|-------|
> | Soft prompt with our loss | 88.2   | 43.4  | 66.6  | 89.6   | 69.2  | 54.0      | 89.6    | 44.2  | 85.2  |
> | Soft‑prompt               | 84.68  | 38.40 | 75.68 | 84.96  | 73.28 | 62.72     | 82.88   | 55.32 | 75.76 |
> | **Ours**                  | **97.08** | **58.32** | **89.06** | **91.84** | **92.64** | **89.47** | **97.92** | **79.24** | **94.56** |
>
> > Question 1: Application to SFT and PEFT algorithm.
>
> As noted in our response to Weakness 1, while the proposed regularization terms are compatible with any SFT or PEFT method, these methods are generally tailored for fine-tuning in data-rich scenarios. In contrast, our approach is specifically designed for few-shot adaptation, where methods like SFT and LoRA are prone to overfitting.
>
> > Weakness 2 and Question 2: The effects of different numbers of exemplars.
>
> We conduct few-shot learning with 5 samples per category. To explore the effects of different numbers of samples, we conduct experiments on various models, Llama2 and Llama3-Instruct, and datasets, HateSP18 and Subj, with the experiment results below. In general, our method benefits from the increasing number of samples, but the model’s performance saturates after 15 shots.
>
> ### Llama2-7B
>
> | HateSP18 | 3-shot | 5-shot | 10-shot | 15-shot | 20-shot |
> |----------|--------|--------|---------|---------|---------|
> | ICL      | 68.81  | 70.24  | 70.64   | 74.23   | 70.72   |
> | Ours     | 77.63  | 83.04  | 83.73   | 86.09   | 84.97   |
>
>
> ### Llama3-Instruct
>
> | Subj | 3-shot | 5-shot | 10-shot | 15-shot | 20-shot |
> |-------|--------|--------|---------|---------|---------|
> | ICL   | 63.76  | 57.48  | 54.73   | 61.80   | 58.32   |
> | Ours  | 88.76  | 92.64  | 93.62   | 95.40   | 95.23   |
>
> > Question 3: Hyperparameter sensitivity analysis
>
> We provide details of the grid search over the hyperparameters $\lambda_1$​ and $\lambda_2$ on the Emoc and Hatesp18 datasets. The results demonstrate that our proposed loss function is robust to variations in these hyperparameters within the moderate range of $[10^{-5}, 10^{-3}]$, with only limited performance fluctuations. Accordingly, we set $\lambda_1$ and $\lambda_2$​ within this range for all datasets in our experiments. Therefore, this confirms that the range $[10^{-5}, 10^{-3}]$ is effective and generalizes well across all experimental settings.
>
> ### EmoC
> | $\lambda_1$ \  $\lambda_2$ | 0.00001 | 0.0001 | 0.001 | 0.01  | 0.1   |
> |----------------------------|---------|--------|--------|--------|--------|
> | 0.00001                   | 77.15   | 77.58  | 76.19  | 77.47  | 77.30  |
> | 0.0001                    | 78.05   | 78.49  | 78.71  | 77.52  | 74.11  |
> | 0.001                     | 78.09   | **79.24** | 78.31  | 76.27  | 76.09  |
> | 0.01                      | 77.66   | 78.89  | 79.38  | 78.29  | 75.68  |
> | 0.1                       | 75.77   | 74.31  | 74.91  | 74.88  | 74.22  |
>
> ### HateSp18
>
> | $\lambda_1$ \ $\lambda_2$ | 0.00001 | 0.0001 | 0.001 | 0.01  | 0.1   |
> |---------------------------|---------|--------|--------|--------|--------|
> | 0.00001                   | 87.58   | 88.06  | 88.97  | 87.43  | 85.45  |
> | 0.0001                    | 88.25   | **89.47** | 88.95  | 88.11  | 85.09  |
> | 0.001                     | 87.89   | 88.71  | 89.30  | 87.88  | 85.10  |
> | 0.01                      | 87.53   | 88.17  | 88.61  | 87.01  | 85.49  |
> | 0.1                       | 87.15   | 86.75  | 87.88  | 86.03  | 86.20  |

---

> > ### Author Response · Authors · 2025-08-04
> >
> > Dear Reviewer,
> >
> > We appreciate the time and effort you have taken to review our submission and provide valuable feedback. We have carefully addressed the comments raised during the first round and submitted our responses accordingly. As the discussion period is ending soon, we would greatly appreciate any further feedback you might have. Please let us know if there are any remaining concerns or questions, so we can address them in time.
> >
> > Thank you again for reviewing our paper.
> >
> > Best regards,
> >
> > Authors

---

> > > ### Comment · Reviewer_CPuh · 2025-08-05
> > > **Response from reviewer CPuh**
> > >
> > > I'd like to thank the authors for clarifying the differences with soft prompt, lora, and the additional experiments on increasing the number of samples. My concerns are resolved.

---

> > > > ### Author Response · Authors · 2025-08-05
> > > >
> > > > Thank you very much for your support and feedback. We are glad to know that we have addressed your comments.

---

### Official Review · Reviewer_HYpd · 2025-07-03

**Clarity:** 2
**Significance:** 2
**Originality:** 4
**Rating:** 5
**Confidence:** 3

**Summary:**

This paper proposes Optimization-Inspired Few-Shot Adaptation (OFA), an approach for adapting LLMs to new tasks using only a few demonstration examples, addressing key limitations in existing few-shot adaptation methods. The paper builds upon previous finding which view the forward pass of LLMs as gradient descent optimization. The key insight is that LayerNorm parameters can be finetuned per task to learn preconditioning matrices for each gradient descent step, which happens during the forward pass. The method introduces two objective terms: one for optimization efficiency (step-ratio minimization) and another for generalization (sharpness minimization via Hutchinson approximation). The authors evaluate OFA across multiple classification tasks and LLM architectures, demonstrating consistent improvements over strong baselines.

**Questions:**

- Could the authors comment on the sensitivity of the OFA hyperparameters?
- What is the overall cost of finetuning an LLM with OFA? How does it compare to the baselines?
- In the transition from equation L144 to the preconditioning matrix formulation (equation L147), a bias term $\frac{\mu_t}{\sigma_t}$ appears to be dropped. Could the authors elaborate on this step?

**Ethical Concerns:**

["NO or VERY MINOR ethics concerns only"]

**Final Justification:**

My recommendation for acceptance is based on the novelty and the strong empirical performance of the work. The issues I raised regarding runtimes and hyperparameter sensitivity were addressed during the rebuttal.

**Limitations:**

Yes, but only in Appendix.

**Paper Formatting Concerns:**

No major formatting issues.

**Quality:**

4

**Strengths And Weaknesses:**

### Strengths

Overall, the contribution of this work is solid.

(Novelty): To the best of my knowledge, this line of research is novel. The authors rely on optimization and in-context learning theory to guide their design choices. Importantly, the paper also includes compelling empirical evidence that is consistent with what theory predicts.

(Strong empirical performance): OFA demonstrates clear improvements over baselines across multiple classification datasets and LLM architectures.

### Weaknesses

The practical usability of the approach is still unclear to me since the paper lacks some details about the experimental validation.

(Lack of hyperparameter ablation): The method introduces two additional hyperparameters (λ₁, λ₂) requiring grid search across a substantial range (6 different values each), potentially making the method less practical and adding tuning burden. An ablation study over the two hyperparameters seems crucial to me. It would be informative to clarify whether the values of these hyperparameters are specific for each experiment or shared across all.

(Lack of details about the training setup / overhead): Another missing detail is the runtime during training/finetuning. It would be interesting to get insights on the difficulty of the optimization when considering the two loss terms.

---

> ### Author Rebuttal · Authors · 2025-07-29
>
> > Question 1: hyperparameter sensitivity of our algorithm, OFA.
>
> Certainly. We provide details of the grid search over the hyperparameters $\lambda_1$​ and $\lambda_2$ on the Emoc and Hatesp18 datasets. The results demonstrate that our proposed loss function is robust to variations in these hyperparameters within the moderate range of $[10^{-5}, 10^{-3}]$, with only limited performance fluctuations. Accordingly, we set $\lambda_1$ and $\lambda_2$​ within this range for all datasets in our experiments. Therefore, this confirms that the range $[10^{-5}, 10^{-3}]$ is effective and generalizes well across all experimental settings.
>
> ### EmoC
>
> | $\lambda_1$ \  $\lambda_2$ | 0.00001 | 0.0001 | 0.001 | 0.01  | 0.1   |
> |----------------------------|---------|--------|--------|--------|--------|
> | 0.00001                   | 77.15   | 77.58  | 76.19  | 77.47  | 77.30  |
> | 0.0001                    | 78.05   | 78.49  | 78.71  | 77.52  | 74.11  |
> | 0.001                     | 78.09   | **79.24** | 78.31  | 76.27  | 76.09  |
> | 0.01                      | 77.66   | 78.89  | 79.38  | 78.29  | 75.68  |
> | 0.1                       | 75.77   | 74.31  | 74.91  | 74.88  | 74.22  |
>
> ### HateSp18
>
> | $\lambda_1$ \ $\lambda_2$ | 0.00001 | 0.0001 | 0.001 | 0.01  | 0.1   |
> |---------------------------|---------|--------|--------|--------|--------|
> | 0.00001                   | 87.58   | 88.06  | 88.97  | 87.43  | 85.45  |
> | 0.0001                    | 88.25   | **89.47** | 88.95  | 88.11  | 85.09  |
> | 0.001                     | 87.89   | 88.71  | 89.30  | 87.88  | 85.10  |
> | 0.01                      | 87.53   | 88.17  | 88.61  | 87.01  | 85.49  |
> | 0.1                       | 87.15   | 86.75  | 87.88  | 86.03  | 86.20  |
>
>
> > Question 2:  Training cost of our algorithm?
>
> We compare the training costs of various algorithms on DBPedia, the dataset with the largest number of categories in our experimental suite, using LLaMA3-8B-Instruct.
> Compared to LoRA-based models, our method achieves faster training times and better performance, while maintaining the same inference cost. When compared with embedding-based approaches such as Label-Anchor, Task-Vector, and I2CL, our method incurs slightly higher training time but achieves significantly better performance during inference, along with reduced inference time. Finally, in comparison to IA3 and Soft-Prompt, our method demonstrates substantial improvements in both inference efficiency and accuracy.
>
> | Method           | Soft-prompt | Label-anchor | Task-vector | IA3     | I2CL    | LoRA (Rank16) | LoRA (Rank1) | Ours    |
> |------------------|-------------|---------------|--------------|---------|---------|---------------|----------------|---------|
> | Training cost (s) | 196.08      | 50.41         | 213.25       | 324.12  | 312.50  | 483.52       |  449.23       | 367.05  |
> | Inference cost (s) | 53.64       | 52.41         | 56.78        | 52.40   | 52.59   | 51.38         | 51.35          | 51.37   |
> |Performance (Accuracy %)| 82.88 | 87.48| 79.52 | 92.56 | 91.20 | 92.62 | 57.88  | 97.92 |
>
> > Question 3: The omitted bias term.
>
> We appreciate this issue being identified by the reviewer. In the mainstream open-source LLMs, such as Llama, BLOOM, Falcon, Qwen, and MPT, a special type of LayerNorm, RMSNorm, is usually used, with the definition as: $\text{RMSNorm}(x) =  \gamma \cdot \frac{x}{\sqrt{\frac{1}{d} \sum_{i=1}^d x_i^2 + \epsilon}} $ where $\gamma$ is learnable parameter. In this way, the computation of RMSNorm can be interpreted by setting the mean  $\mu$ in LayerNorm to 0. As a result, the bias term is removed by multiplying by 0. In the normal LayerNorm case, such as GPT2-XL, the bias term can be bounded by a constant; then omitting this will simplify our analysis without affecting the conclusion that optimizing our proposed loss leads to faster convergence. To further test whether our algorithm is robust to the normal layer norm, the GPT2-XL (with LayerNorm) trained with our algorithm on the few-shot learning tasks also demonstrates superior performance over all the baseline algorithms in Table 4 of our submission. Therefore, we omit this bias to highlight the main focus, $P_t \nabla\mathcal{L}(Z_t)$, of our work. We will explain these reasons clearly in our revised version.
>
> ### Few-shot performance of GPT2-XL. We quote the experiment results below from Table 4 in the appendix of our original submission.
>
> | Dataset / Method   | SST-2            | SST-5            | TREC             | Subj             | HateSp18         | EmoC             | MR               |
> |--------------------|------------------|------------------|------------------|------------------|------------------|------------------|------------------|
> | Zero-shot          | 74.76            | 30.44            | 35.40            | 64.88            | 70.84            | 37.88            | 71.36            |
> | Few-shot (ICL)     | 73.65 $\pm$ 8.89 | 35.95 $\pm$ 2.39 | 60.64 $\pm$ 5.00 | 63.82 $\pm$ 10.55| 51.86 $\pm$ 3.22 | 38.62 $\pm$ 6.87 | 75.79 $\pm$ 9.25 |
> | Soft-prompt        | 61.04 $\pm$ 3.45 | 23.96 $\pm$ 2.09 | 40.60 $\pm$ 10.15| 55.44 $\pm$ 4.12 | 63.92 $\pm$ 7.06 | 36.68 $\pm$ 2.70 | 57.60 $\pm$ 3.53 |
> | Label-anchor       | 63.40 $\pm$ 8.82 | 22.36 $\pm$ 3.37 | 66.36 $\pm$ 10.69| 55.56 $\pm$ 4.26 | 54.88 $\pm$ 4.53 | 36.68 $\pm$ 2.70 | 60.20 $\pm$ 3.32 |
> | Task-vector        | 81.08 $\pm$ 4.87 | 28.52 $\pm$ 1.37 | 41.40 $\pm$ 5.35 | 71.81 $\pm$ 1.86 | 62.48 $\pm$ 2.83 | 37.60 $\pm$ 2.48 | 78.40 $\pm$ 2.26 |
> | IA3                | 86.64 $\pm$ 2.89 | 40.52 $\pm$ 2.25 | 70.96 $\pm$ 8.61 | 71.52 $\pm$ 8.46 | 70.84 $\pm$ 3.63 | 62.24 $\pm$ 3.50 | 83.24 $\pm$ 1.09 |
> | I2CL               | 80.16 $\pm$ 3.98 | 35.04 $\pm$ 2.60 | 51.48 $\pm$ 5.26 | 65.96 $\pm$ 4.83 | 68.32 $\pm$ 4.76 | 47.92 $\pm$ 1.84 | 83.20 $\pm$ 3.29 |
> | **Ours**           | **88.68 $\pm$ 2.66** | **42.48 $\pm$ 2.51** | **70.60 $\pm$ 6.44** | **86.11 $\pm$ 4.29** | **71.44 $\pm$ 8.65** | **65.30 $\pm$ 4.18** | **84.80 $\pm$ 6.21** |

---

> > ### Author Response · Authors · 2025-08-04
> >
> > Dear Reviewer,
> >
> > We appreciate the time and effort you have taken to review our submission and provide valuable feedback. We have carefully addressed the comments raised during the first round and submitted our responses accordingly. As the discussion period is ending soon, we would greatly appreciate any further feedback you might have. Please let us know if there are any remaining concerns or questions, so we can address them in time.
> >
> > Thank you again for reviewing our paper.
> >
> > Best regards,
> >
> > Authors

---

> > ### Comment · Reviewer_HYpd · 2025-08-04
> >
> > Thanks to the authors for their responses. The answers addressed my comments thus I will maintain my recommendation for acceptance. I suggest the authors to include the additional clarifications / results and discuss the limitations in the main text for the revised version.

---

> > > ### Author Response · Authors · 2025-08-04
> > >
> > > Thank you very much for your support and feedback. We are glad to know that we have addressed your comments. We will revise our paper following your suggestions in the final version.

---

> > > > ### Comment · Reviewer_6QXn · 2025-08-06
> > > >
> > > > It would be helpful to state explicitly that the proposed method applies in both $\rho < 1$ and $\rho > 1$, while noting that the current proof covers only the $\rho <1$ case. Please also define $Z^{*}$ before it is first used. I understand that, given the limited time, the authors do not have the results for tasks other than sentence classification. A brief discussion of the $\rho > 1$ scenario would further strengthen the paper. As the work meets the next acceptance threshold, I have accordingly raised my score.

---

> > > > > ### Author Response · Authors · 2025-08-06
> > > > >
> > > > > Dear Reviewer 6QXn
> > > > >
> > > > > Thank you very much for raising the score, and thank you for your suggestion! We will revise our paper accordingly in the final version.

---

### Official Review · Reviewer_6QXn · 2025-07-03

**Clarity:** 3
**Significance:** 3
**Originality:** 4
**Rating:** 5
**Confidence:** 4

**Summary:**

The paper proposes Optimization-Inspired Few-shot Adaptation (OFA), a lightweight way to dapat a language model with only a small portion of labeled examples. The paper views the transformer block as one step of gradient descent and uses the gradient descent to adapt the model. OFA uniquely adapts an LLM by learning the LayerNorm scales as diagonal pre-conditioners that rescale each layer’s gradient, which is different from the previous optimization-inspired method. Only the scale parameters are fine-tuned. After training, the adapted model runs with zero extra tokens. The experimental results showed that OFA outperforms ICL-based and PEFT-based methods while requiring only fewer parameters and matching zero-shot latency.

**Questions:**

1. Theorem 3.1 requires $\rho_t < 1$ for local contraction. However the paper does not provide and proof or intuition for why we can assume this. Can you provide a theoretical argument or a lemma with conditions on $\eta$ and $P_t$ that help to guarantee this? Or is this possible to supply some empirical evidence on demonstrating $\rho_t < 1$ in most of the cases.

2. Please give a precise definition of $Z^*$ and state where the symbol first appears.

3. Did the paper look at tasks other than single-sentence classification? Does the method work for other tasks, such as generation, or name entity recognition?

4. When do the authors plan to release the implementation and pre trained checkpoints for reproducibility?

**Ethical Concerns:**

["NO or VERY MINOR ethics concerns only"]

**Final Justification:**

The authors explained that their methods works for the case when $\rho >1$.  This makes the work more general. It would be helpful to state explicitly that the proposed method applies in both $\rho < 1$ and $\rho > 1$ in the paper, while noting that the current proof covers only the $\rho <1$ case. As the work meets the next acceptance threshold, I have accordingly raised my score.

**Limitations:**

Yes

**Paper Formatting Concerns:**

No paper formatting concerns.

**Quality:**

3

**Strengths And Weaknesses:**

For quality, the paper conducted thorough experiments comparing OFA with ICL-based and PEFT-based methods and other optimization-inspired methods. The tasks however are limited to single sentence classification, no generation, or name entity recognition. For clarity, the paper explains clearly how OFA is compared to ICL and PEFT, and the algorithm and figures are easy to follow. The work showed that tweaking only LayerNorm scales can beat ICL and PEFT. Offer practical insights for the community. It is also the first to turn the LayerNorm scales into a preconditioner and bridges the optimization-inspired methods with PEFT. Other weaknesses are discussed in the Questions section.

---

> ### Author Rebuttal · Authors · 2025-07-29
>
> > Question 1: Local contraction with $\rho_t < 1$. Can the author justify this assumption?
>
> Our work is motivated by an optimization perspective, where the assumption $\rho < 1$ implies that the optimization process converges [1,2]. In the in-context learning framework introduced in Section 3.1, the forward pass of the LLM can be interpreted as an optimization process, with each attention layer corresponding to one step of gradient descent. Our analysis relies on this assumption, as a rigorous proof that optimizing our loss leads to fast convergence is only feasible under the convergence condition $\rho < 1$. Specifically, the preconditioning matrix must satisfy $\rho = || I - \eta P_t H || < 1$.
>
> Conversely, if this assumption does not hold, the internal optimization implemented by the forward LLM pass cannot be strictly interpreted as an optimization process. Nevertheless, our algorithm is designed to find the optimal task-specific $\rho$ by searching for the corresponding preconditioning matrices $P_t$​, aiming to accelerate convergence regardless of whether $\rho$ is initially greater or less than 1. In other words, $\rho < 1$ is not a strict constraint for our algorithm. Instead, our method applies to both $\rho < 1$ and $\rho > 1$ scenarios, with the goal of optimizing $\rho$ rather than requiring it.
>
> In Figure 3 of our submission, one can observe that our algorithm produces the lowest $\rho$ among all the baseline models, which implies the fastest convergence speed.
>
> >  Question 2: Definition of $Z^{*}$
>
> We clarify that $Z^{\*}$ denotes the optimum of the objective function, defined as $Z^{\*}= \arg\min_{Z} \mathcal{L}$, where $\mathcal{L}$ represents the objective of the internal optimization performed during the forward pass of the LLM under the in-context learning setting. This optimization occurs between lines 131 and 132 of our submission. $Z^{\*}$ is first introduced between lines 156 and 157. We sincerely thank the reviewer for the helpful feedback and will ensure this clarification is explicitly highlighted in the revised version.
>
>
> > Question 3: Can the proposed algorithm work on the few-shot generalization task and name entity recognition?
>
> We sincerely thank the reviewer for the insightful suggestion to evaluate our algorithm on few-shot generation and named entity recognition (NER) tasks. Our work follows in the mainstream in-context learning setting [3, 4, 5, 6, 7], where sentence classification serves as the main benchmark for evaluation. Furthermore, our current experiments already encompass a diverse set of models, such as Llama2-7B, Llama3-8B, Llama3-8B-Instruct, and GPT2-XL, and datasets, SST-2, SST-5, TREC, AGNews, Subj, HateSp18, DBPedia, EmoC, and MR. While it is theoretically possible to extend our method to few-shot generation and NER, doing so would entail significant methodological modifications and implementation effort for both our approach and the corresponding baselines, which would be difficult to complete in the rebuttal phase given the the limited time available. We agree that these directions are valuable extension work, and we intend to explore them more thoroughly in future work.
>
> > Question 4: Releasing code and model
>
> Certainly, we will release our code and trained models once our paper is accepted, to support implementation transparency and ensure reproducibility.
>
> [1] Nesterov, Yurii. Lectures on convex optimization. Vol. 137. Berlin: Springer International Publishing, 2018.
>
> [2] Boyd, Stephen P., and Lieven Vandenberghe. Convex optimization. Cambridge University Press, 2004.
>
> [3] Huang, Yu, Yuan Cheng, and Yingbin Liang. "In-context Convergence of Transformers. In ICML. 2024.
>
> [4] Bai, Yu, et al. "Transformers as statisticians: Provable in-context learning with in-context algorithm selection." In NeurIPS. 2023.
>
> [5] Ahn, Kwangjun, et al. "Transformers learn to implement preconditioned gradient descent for in-context learning." In NeurIPS. 2023.
>
> [6] Akyürek, Ekin, et al. "​​ What learning algorithm is in-context learning? Investigations with linear models." In ICLR. 2023.
>
> [7] Li, Zhuowei, et al. "Implicit In-context Learning." In ICLR. 2025

---

> > ### Comment · Reviewer_6QXn · 2025-08-04
> >
> > Thank the authors for the detailed clarification, I have updated my final justification and scores.

---

> > > ### Author Response · Authors · 2025-08-04
> > >
> > > Thank you very much for your support and feedback. We are glad to know that we have addressed your comments. If you have any further concerns or suggestions regarding our submission, we will try our best to address them.

---

> ### Author Response · Authors · 2025-08-04
>
> Dear Reviewer,
>
> We appreciate the time and effort you have taken to review our submission and provide valuable feedback. We have carefully addressed the comments raised during the first round and submitted our responses accordingly. As the discussion period is ending soon, we would greatly appreciate any further feedback you might have. Please let us know if there are any remaining concerns or questions, so we can address them in time.
>
> Thank you again for reviewing our paper.
>
> Best regards,
>
> Authors

---

### Note · Authors · 2025-08-15

Dear all,

We sincerely appreciate the time and effort the reviewers dedicated to evaluating our paper, and we are grateful for their constructive feedback, which we found both insightful and valuable. We are encouraged that our work received all positive feedback, with reviewers 6QXn and 7e2o promising to raise their scores. It is gratifying that all reviewers recognised our method as both novel and concrete, and agreed that their concerns were fully addressed during the rebuttal process.

We thank the reviewers once again for their valuable feedback and suggestions. We will incorporate these insights to improve the final version of our paper further and remain open to any additional input.

Best regards,

Authors

---

### Decision · Program_Chairs · 2025-09-17

**Decision:**

Accept (spotlight)

**Comment:**

Based on my reading the reviews, author responses and discussion, I see the following strengths of this manuscript:

- **S1.** Based on an insightful connection between optimization and the transformer forward pass, the submission proposes a novel and unique low-parameter LLM adaptation for few-shot problems, and provides a strong and elaborate empirical evaluation (multiple datasets, architectures).

- **S2.** Along with the parameterization, the submission also proposes novel loss terms such as step-size ratio and sharpness regularization, which significantly improve the performance on few-shot problems. Furthermore, these loss can also be applied to other PEFT and LoRA based adaptation schemes though these variations of the baselines still underperform the proposed scheme.

I also saw various limitations such as (i) unknown sensitivity to new hyperparameters for the new loss terms, (ii) additional computation time due to the layer-wise sharpness estimation, and (iii) the additional smoothness assumptions in the analysis. However, it seems to me that all the limitations are somewhat minor, and adequately addressed in the discussion with additional empirical evaluations (for the hyperparameter dependencies and runtime comparisons).

Based on this, I am recommending an accept for this paper.